# Brain Mapping with Dense Features: Grounding Cortical Semantic Selectivity in Natural Images With Vision Transformers

**Andrew F. Luo**[1,2]        **Jacob Yeung**[1]        **Rushikesh Zawar**[1]        **Shaurya Dewan**[1]

**Margaret M. Henderson**[1]        **Leila Wehbe\***[1]        **Michael J. Tarr\***[1]

[1] Carnegie Mellon University        [2] The University of Hong Kong
aluo@hku.hk, {jacobyeung,rzawar,srdewan,mmhender,
lwehbe,michaeltarr}@cmu.edu

## Abstract

We introduce BrainSAIL (**S**emantic **A**ttribution and **I**mage **L**ocalization), a method for linking neural selectivity with spatially distributed semantic visual concepts in natural scenes. BrainSAIL leverages recent advances in large-scale artificial neural networks, using them to provide insights into the functional topology of the brain. To overcome the challenge presented by the co-occurrence of multiple categories in natural images, BrainSAIL exploits semantically consistent, dense spatial features from pre-trained vision models, building upon their demonstrated ability to robustly predict neural activity. This method derives clean, spatially dense embeddings without requiring any additional training, and employs a novel denoising process that leverages the semantic consistency of images under random augmentations. By unifying the space of whole-image embeddings and dense visual features and then applying voxel-wise encoding models to these features, we enable the identification of specific subregions of each image which drive selectivity patterns in different areas of the higher visual cortex. This provides a powerful tool for dissecting the neural mechanisms that underlie semantic visual processing for natural images. We validate BrainSAIL on cortical regions with known category selectivity, demonstrating its ability to accurately localize and disentangle selectivity to diverse visual concepts. Next, we demonstrate BrainSAIL's ability to characterize high-level visual selectivity to scene properties and low-level visual features such as depth, luminance, and saturation, providing insights into the encoding of complex visual information. Finally, we use BrainSAIL to directly compare the feature selectivity of different brain encoding models across different regions of interest in visual cortex. Our innovative method paves the way for significant advances in mapping and decomposing high-level visual representations in the human brain.

## 1 Introduction

Understanding how the human brain processes and represents visual information from natural experience is a fundamental challenge in neuroscience. The vast majority of our knowledge of the visual system comes from tightly controlled experiments using simplified, hand-crafted images or, at best, real-world photographs of objects against noise backgrounds. Although this paradigm has revealed a pattern of preferential neural responses to semantic categories such as faces, places, bodies, words, objects, and food (Sergent et al., 1992; Allison et al., 1994; McCarthy et al., 1997; Kanwisher et al., 1997; Aguirre et al., 1996; Epstein & Kanwisher, 1998; Downing et al., 2001; Grill-Spector, 2003; Malach et al., 1995; Khosla et al., 2022; Pennock et al., 2023; Jain et al., 2023), the visual world we actually experience consists of rich, complex scenes containing many co-occurring objects, textures, and contextual associations (Simoncelli & Olshausen, 2001; Torralba & Oliva, 2003). As such, using minimal or single-object stimuli narrows the space of hypothesis testing and limits the ecological relevance of any conclusions, leaving us with an incomplete characterization of how the brain represents and processes real-world visual stimuli.

---

\* Co-corresponding authors.

Recent developments in computer vision models trained on web-scale datasets have enabled learning rich multimodal representations that capture semantic concepts in a human-aligned manner (Conwell et al., 2022; Wang et al., 2023a). In this work, we introduce a novel methodology that leverages the power of such models to decompose selectivity patterns in visual cortex by analyzing responses to dense, localized semantic features present in naturalistic images: **S**emantic **A**ttribution and **I**mage **L**ocalization ("BrainSAIL"). BrainSAIL allows us to isolate the specific image regions that activate different cortical areas when viewing naturalistic scenes. This method allows us to focus on selectivity within complex naturalistic images, thereby enabling a richer decomposition grounded in the full semantic complexity of natural visual experiences.

The core of BrainSAIL involves extracting spatially dense semantic embeddings from images using state-of-the-art models such as CLIP, DINO, or SigLIP (Radford et al., 2021; Caron et al., 2021; Zhai et al., 2023). These embeddings bridge the traditionally disparate domains of raw vision data, dense deep semantic features, and measured neural responses. Within this rich embedding space, we can isolate and identify the specific visual features and corresponding image regions that drive selectivity effects in different cortical areas during perception of naturalistic visual scenes. By concurrently modeling localized semantic information, high-level semantic categories, and observed brain activity patterns, BrainSAIL can tease apart the image-level visual drivers of neural tuning preferences across higher visual areas. We validate this dense feature mapping method on a large-scale fMRI dataset consisting of human participants viewing many thousands of diverse natural images that span a wide range of semantic categories and visual statistics (Allen et al., 2022).

BrainSAIL's dense embedding framework offers an interpretable view of feature representations across visual regions of the brain. Critically, this view explicitly grounds neural selectivity to localized semantic characteristics inherent in real-world visual experiences. First, we demonstrate the utility of our model for natural images applied to known category-selective regions of the cortex. Second, we show that our model can be used to identify the preference of brain regions sensitive to scene statistics. Finally, we use our model to compare and contrast the feature selectivity for different vision foundation models. In sum, the dense semantic grounding realized in BrainSAIL enables exciting new directions towards understanding and modeling high-level visual representation in humans.

## 2 Related Work

A growing body of work leveraging computational modeling and machine learning has explored semantic representation in the higher visual cortex. Approaches include generative image models (Ratan Murty et al., 2021; Gu et al., 2022; Pierzchlewicz et al., 2023; Luo et al., 2023; 2024) and the decoding of visual stimuli (Takagi & Nishimoto, 2022; Chen et al., 2022; Doerig et al., 2022; Ferrante et al., 2023; Liu et al., 2023; Scotti et al., 2024; Yeung et al., 2024). These diverse studies are united by their consideration of the stimulus image as a whole, primarily focusing on the global information contained within the image rather than the individual scene components. In contrast, the method we introduce decomposes an image into its semantic components, enabling the identification of individual, semantically meaningful activating concepts within complex natural images.

**Semantic Representation in the Visual Cortex.** Using hand-crafted image stimuli, functional mapping studies have identified regions in the human brain that respond preferentially to stimuli representing distinct semantic concepts such as faces, places, bodies, words, objects, and food (Desimone et al., 1984; Sergent et al., 1992; Allison et al., 1994; McCarthy et al., 1997; Kanwisher et al., 1997; Gauthier & Tarr, 1997; Aguirre et al., 1996; Epstein & Kanwisher, 1998; Aguirre et al., 1998; O'Craven & Kanwisher, 2000; Nakamura et al., 2000; Aminoff et al., 2007; Downing et al., 2001; Cohen et al., 2000; Grill-Spector, 2003; Khosla et al., 2022; Pennock et al., 2023; Jain et al., 2023). One limitation of this simplified approach is that it may not fully capture the contextual complexity of natural vision (Gallant et al., 1998; Mahon, 2022). Addressing this concern, recent work on image-computable encoders has enabled computational tests of visual selectivity using naturalistic images (Naselaris et al., 2011; Huth et al., 2012; Yamins et al., 2014; Eickenberg et al., 2017; Wen et al., 2018; Kubilius et al., 2019; Popham et al., 2021; Conwell et al., 2022; Wang et al., 2023a; Luo et al., 2023; Prince et al., 2023; Adeli et al., 2023; Luo et al., 2024; Yang et al., 2024b;a; Efird et al., 2024). Building on this work, our method leverages state-of-the-art brain encoding backbones based on vision transformers (Dosovitskiy et al., 2020; Wang et al., 2023a) to further explore finer-grained semantic representation in visual cortex.

**Visual Contrastive Representation Learning.** Self- or weakly-supervised vision models that use contrastive (Xing et al., 2002; Schultz & Joachims, 2003; Chopra et al., 2005; Sohn, 2016; Wu et al.,

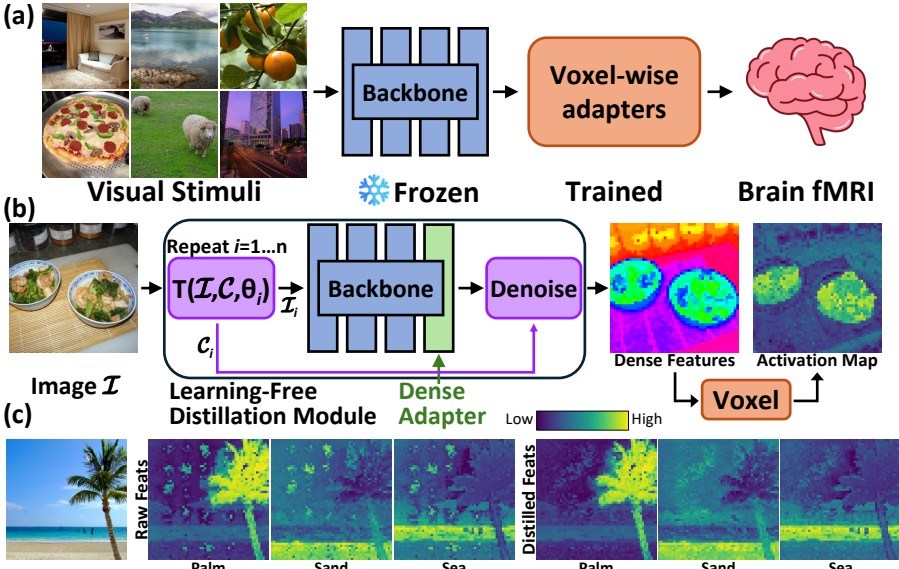

Figure 1: **The BrainSAIL framework leverages dense visual features**. **(a)** An fMRI encoder learns a map from images to voxel-wise activations in the brain. Encoders leveraging frozen foundation models based on vision transformers (ViTs) with voxel-wise adapters are currently the highest accuracy models for brain prediction (Conwell et al., 2022; Wang et al., 2023a). **(b)** Given an image and a ViT backbone for the fMRI encoder, we modify the backbone to output dense features. The dense backbone is wrapped inside of a **Learning-Free Distillation Module**. This module takes an image $\mathcal{I}$ and 2D image coordinates $\mathcal{C}$, and generates transformed images and coordinates $(\mathcal{I}_i, \mathcal{C}_i)$ for a given transform $\theta_i$. The dense features and transformed coordinates are provided to a denoising module to generate clean dense features. The frozen voxel-wise adapter from (a) is applied to each patch to generate dense relevance maps which highlight the image regions activating the voxel. **(c)** Using `CLIP ViT-B/16` with the latest `NACLIP` adapter, we show relevance maps using the CLIP text encoder. The NACLIP raw features are highly noisy and contain artifacts, while the distilled features are localized to the relevant semantic components with high accuracy. Note that we achieve state-of-the-art open vocabulary CLIP-based segmentation results using our method.

2018; Musgrave et al., 2020) and masked prediction objectives (Pathak et al., 2016; Kolesnikov et al., 2019; Chen et al., 2020; Zhao et al., 2021; Li et al., 2021; Zhou et al., 2021) are scalable and can be trained on massive, diverse datasets to achieve high zero-shot performance on downstream tasks. Contrastive models such as CLIP, DINO, and SigLIP demonstrate strong classification performance without further fine-tuning (Radford et al., 2021; Caron et al., 2021; Oquab et al., 2023; Zhai et al., 2023). Models that jointly train on language and vision (CLIP/SigLIP) can also classify images using text-based descriptions without fine-tuning. Interestingly, this high level of performance is mirrored in the fact that contrastive models show high performance for predicting neural responses in visual cortex when paired with linear probes (Conwell et al., 2022; Wang et al., 2023a).

**Exploring the Brain with Foundation Models.**    There has been strong interest in leveraging generative models for decoding (reconstructing) visual stimuli conditioned on brain activations (Kamitani & Tong, 2005; Han et al., 2019; Seeliger et al., 2018; Shen et al., 2019; Ren et al., 2021; Takagi & Nishimoto, 2022; Chen et al., 2023; Lu et al., 2023; Ozcelik & VanRullen, 2023; Doerig et al., 2022; Ferrante et al., 2023; Liu et al., 2023; Mai & Zhang, 2023; Scotti et al., 2024). A related approach generates novel stimuli that are posited to best to activate a target brain region (as opposed to reconstructing the original stimulus) (Walker et al., 2019; Bashivan et al., 2019) with recent attempts utilizing GANs or Diffusion models to constrain the synthesized output (Ponce et al., 2019; Ratan Murty et al., 2021; Gu et al., 2022; Luo et al., 2023; 2024). While these models have shown positive results, they all rely on images as a whole, whereas BrainSAIL seeks to disentangle complex images into their semantically meaningful components and localize those parts of the image that elicit activation for different brain voxels or regions.

## 3    METHODS

Our aim is to generate spatial attribution maps for arbitrary voxels in the higher visual cortex. Unlike the early visual cortex, which is believed to be primarily selective for "simple features" (Stork &

Wilson, 1990), the higher visual cortex exhibits semantic selectivity – a pattern that, at present, is best predicted by deep networks (Conwell et al., 2022; Wang et al., 2023a). As illustrated in Figure 1, to create spatial attributions maps for brain voxels, we first train voxel-wise fMRI encoders to map images to brain activations. Second, we derive dense features from pre-trained vision transformers (ViT) used as the backbone for these encoders. Third, we demonstrate that an artifact-free dense feature map can be derived for high-throughput exploration of selectivity with the visual cortex.

## 3.1 Image-to-Brain Encoders for the Higher Visual Cortex

A voxel-wise image-computable fMRI encoder is a model $F_\phi$ that predicts fMRI activations (betas) for $B \in \mathbb{R}^{1 \times N}$ where $N$ represents the number of voxels in the brain. The encoder is conditioned on image input $\mathcal{I} \in \mathbb{R}^{H \times W \times 3}$, where $F_\phi(\mathcal{I}) \Rightarrow B$. Recent work has demonstrated that encoders that rely on features extracted from large vision foundation models achieve excellent predictive performance, where higher visual cortex is best predicted by deeper layers in the model (Wang et al., 2023a). In this setting, the backbone model is usually frozen, while a per-voxel adapter typically parameterized as a linear layer is trained to map from network features to voxel activations. In that we focus on the higher visual cortex exclusively, we utilize a two component design for our encoder: (1) a frozen vision foundation model backbone $G(\mathcal{I})$ which outputs a $R^{1 \times M}$ dimension embedding vector for each image; (2) a per-voxel adapter parameterized as a linear probe with weight $W \in \mathcal{R}^{M \times N}$ and bias $b \in \mathcal{R}^{1 \times N}$, which takes as input a unit-norm image embedding.

$$\left[ \frac{G_{\text{img}}(\mathcal{I})}{\|G_{\text{img}}(\mathcal{I})\|_2} \times W + b \right] \Rightarrow B \tag{1}$$

It should be noted that BrainSAIL is not restricted to linear probes, and can work with arbitrary voxel-wise parameterizations, including MLPs. Linear probes are used here as they are widely adopted in fMRI encoder literature and empirically achieve good performance. BrainSAIL is compatible with any Vision Transformer (ViT)-based model, making it readily applicable to the vast majority of modern visual foundation models which predominantly employ ViT architectures. Additional results are presented in the supplemental. We train our model with MSE loss, and evaluate the encoder on the test set. In Figure 6 we show that our encoder achieves state-of-the-art $R^2$.

## 3.2 Deriving Dense Features from ViT backbones

The emergence of vision models trained on a contrastive image-text objective has fueled interest in zero-shot open-vocabulary image classification methods. For example, CLIP has shown that images can be classified without foreknowledge of the test time classes during training; instead the category of interest can be described using language during test time. Of late, this capability has been extended from classification to segmentation. Compared to methods that require human annotation (Li et al., 2022) and perform poorly on out-of-distribution images (Jatavallab-hula et al., 2023; Kerr et al., 2023), these new methods require no further training and directly extract dense features that lie in the same space as the image/text embedding. These dense feature extraction methods operate by modifying the last self-attention (SA) block within the typical ViT architecture (MaskCLIP, Zhou et al. (2022); SCLIP, Wang et al. (2023b); NACLIP, Hajimiri et al. (2024)). For vision models of this sort trained on a contrastive objective, the output is composed of a single `[CLS]` token, which is supervised using a contrastive loss; and numerous patch tokens which correspond to specific spatial locations. Let $(q_i, k_i, v_i)$ be the query, key, value features respectively for a single image patch $i$, with a total of $m$ spatial patches. For a given patch $j$ at the final layer, where f denotes any function applied to the `[CLS]` after the last self-attention, the `[CLS]` token and each dense token is a convex combination of $v$ features:

$$\text{Out\_Orig}_j = f\left(\sum_{k=1}^{m}\left[\text{softmax}(\frac{q_j k^T}{C})_j \cdot v_k\right]\right) \quad \text{Out\_Mask}_j = f(v_j) \quad \text{Out\_NA}_j = f\left(\sum_{k=1}^{m}\left[\text{softmax}(\frac{q_j q^T + \omega_j}{C})_j \cdot v_k\right]\right) \tag{2}$$

MaskCLIP directly removes the convex re-weighting and outputs the value feature for each patch token directly. SCLIP and NACLIP reintroduce the weighting to reduce output artifacts, but modify it with correlative self-attention (CSA); or by using CSA with a spatial attentive bias $\omega$. Here, we utilize NACLIP as the dense adaptor for CLIP. The other two backbones in Section 4.4 use an updated ViT architecture with "register tokens" (Darcet et al., 2023). As these have not been explored in the context of CSA, we utilize MaskCLIP as the dense adaptor. We treat dense features as unit-norm.

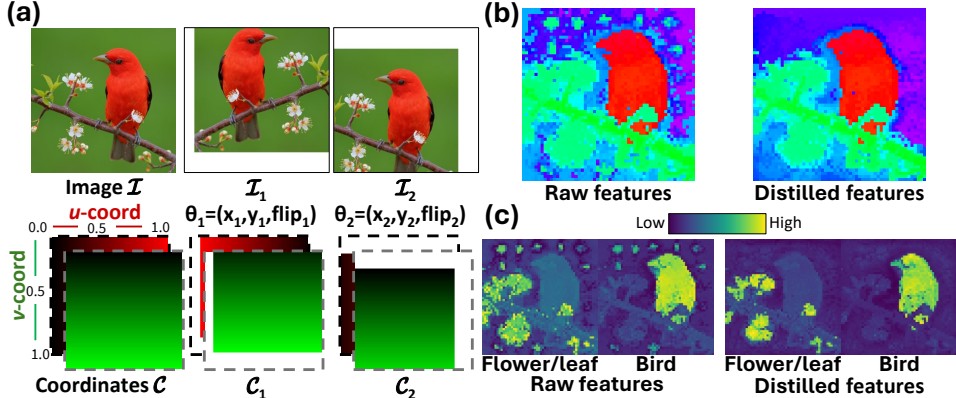

**Figure 2**: **The Learning-Free Distillation Module**. **(a)** Given an image, we generate image-space coordinates $(u, v)$ for each pixel. We then randomly sample from $\theta_{1...n}$, where $\theta_i$ has vertical/horizontal offset, and left-right flips. The augmented images are provided to a frozen backbone with dense adapter. The features are projected to the original image space via an inverse transform $\mathcal{T}^{-1}(\theta_i)$. **(b)** UMAP visualization of the dense features. The same fitted basis is used for both visualizations. **(c)** With CLIP, we can perform zero-shot text queries. Note the artifacts above the bird's head. In practice artifact location is different for each image. The distilled results are significantly better.

### 3.3 LEARNING-FREE FEATURE DISTILLATION

As only the `[CLS]` is supervised in these contrastive models, as shown in Figures 1 and 2, the extracted dense embeddings often have artifacts – even when using the latest NACLIP method which seeks to reduce artifacts. While methods such as Darcet et al. (2023) improve spatial consistency via architectural improvements, they require training the model with architecture modifications that are computationally costly. Methods like Kobayashi et al. (2022) rely on expensive gradient-based optimization using MSE loss over multiple views in $3D$. Consequently, in order to facilitate high-throughput characterization of the visual cortex over large datasets, we propose an efficient learning-free distillation module. Given an image $\mathcal{I}$, we first generate $n$ augmentation parameters $\theta_{1...n}$, where $\theta_i$ consists of a horizontal/vertical offset $(u_i, v_i)$ and horizontal $\text{flip}_i \in \{0, 1\}$. We further generate the image space

---

**Input:** Image $\mathcal{I}$;
        Image space coordinates $\mathcal{C}$;
        Augmentation parameters $\theta_{1...n}$;
        Augmentation function $\mathcal{T}$;
        ViT model with dense adapter $M$;

1. Zero init clean feature tensor $Q$
2. Zero init count tensor $K$
3. **For** i in $\{1...n\}$:
4.     $\theta_i = (u_i, v_i, \text{flip}_i)$
5.     $(\mathcal{I}_i, \mathcal{C}_i) = \mathcal{T}(\mathcal{I}, \mathcal{C}, \theta_i)$
6.     Dense feature $F_i = M(\mathcal{I}_i)$
7.     $(F_i^{\text{valid}}, \mathcal{C}_i^{\text{valid}}) = \mathcal{T}^{-1}(F_i, \mathcal{C}_i, \theta_i)$
8.     $Q[\mathcal{C}_i^{\text{valid}}] = Q[\mathcal{C}_i^{\text{valid}}] + F_i^{\text{valid}}$
9.     $K[\mathcal{C}_i^{\text{valid}}] = K[\mathcal{C}_i^{\text{valid}}] + 1$
10. **return** $Q/K$

Algo 1: **Learning-Free Feature Distillation**

---

coordinates $\mathcal{C} = (\text{u-coord}, \text{v-coord})$, where $u \in [0, 1]$ goes from left-to-right, while $v \in [0, 1]$ goes top-to-bottom. We describe our full transform in Algorithm 1. These augmentations effectively generate $2D$ "views". Our method distills a clean semantic map, as visual semantics are equivariant to shift and horizontal flips. We note that averaging over the number of augmentation is extracting an *optimal* embedding under mean squared error (squared euclidean). Let $\vec{p^*}$ be the optimal embedding under MSE for a patch, and $\vec{p_i}$ with $i \in \{1...n\}$ be the feature candidates under image augmentation:

$$\vec{p^*} = \min_{\hat{p}} \left( \sum_{i=1}^{n} \|\vec{p_i} - \hat{p}\|_2^2 \right) = \min_{\hat{p}} \left( \|\vec{p_1} - \hat{p}\|_2^2 + ... + \|\vec{p_n} - \hat{p}\|_2^2 \right) \tag{3}$$

$$= \min_{\hat{p}} \left( \vec{p_1}^T \vec{p_1} - 2\vec{p_1}^T \hat{p} + \hat{p}^T \hat{p} + ... + \vec{p_n}^T \vec{p_n} - 2\vec{p_n}^T \hat{p} + \hat{p}^T \hat{p} \right) \quad \text{omitting } \vec{p_i}^T \vec{p_i} \tag{4}$$

$$= \min_{\hat{p}} \left( n \cdot \hat{p}^T \hat{p} - 2 \sum_{i=1}^{n} (\vec{p_i}^T \hat{p}) \right) = \min_{\hat{p}} \left( n \cdot \hat{p}^T \hat{p} - 2n \sum_{i=1}^{n} ((1/n) \cdot \vec{p_i}^T \hat{p}) \right) \tag{5}$$

The objective can be expressed as $\|\hat{p} - (1/n) \sum \vec{p_i}\|_2^2 \geq 0$, then $\vec{p^*} = (1/n) \sum \vec{p_i}$.

In Table 1, we compare pre- and post- smoothing results. Under the Pearson metric, which does not assume prior category knowledge, smoothing yields the best performance in three of four datasets. We apply the voxel-wise adapters to the per patch dense features to derive the final relevance map.

| | ADE20k | | COCO Object | | COCO Stuff | | VOC20 | |
|---|---|---|---|---|---|---|---|---|
| | mIoU↑ | Pearson↑ | mIoU↑ | Pearson↑ | mIoU↑ | Pearson↑ | mIoU↑ | Pearson↑ |
| SCLIP | 16.45 | 0.308 | 33.52 | 0.353 | 21.95 | 0.309 | **81.54** | **0.551** |
| NACLIP | 17.69 | 0.425 | 33.14 | 0.418 | 22.58 | 0.393 | 77.09 | 0.473 |
| +Distilled | **18.19** | **0.443** | **34.15** | **0.435** | **23.08** | **0.405** | 79.09 | 0.489 |

Table 1: **Open-vocabulary segmentation with dense CLIP features.** We validate the effectiveness of our learning-free smoothing approach on segmentation datasets in a zero-shot setting (without any training). This performance is state-of-the-art for open vocabulary segmentation. Note mIoU scores are multiplied by 100. Our smoothing improves the results.

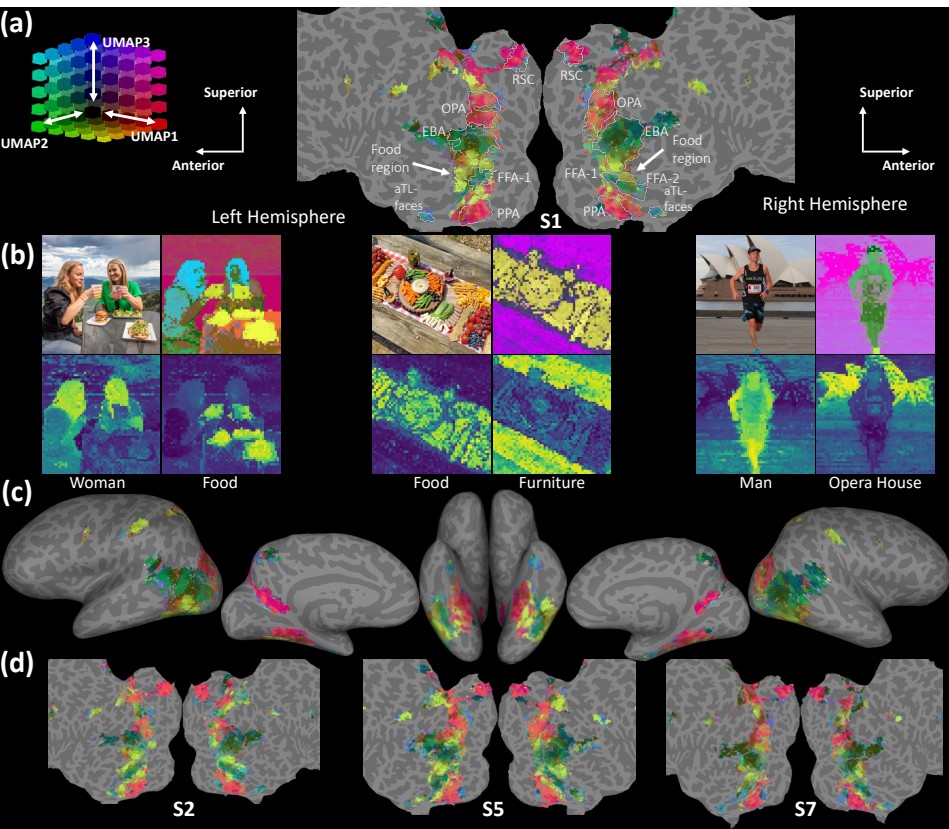

Figure 3: **Joint dimensional reduction of higher visual cortex encoder weights and images using BrainSAIL.** We use a UMAP to perform visualization of the encoder weights. This same UMAP basis is reused for images. **(a)** Cortical flatmap of **S1**. Note that the overlaid white region outlines and labels were derived from *functional localizer data collected independently from the visualized UMAP results*. **(b)** Embeddings from novel images are computed with BrainSAIL and transformed using the fMRI UMAP. For each quartet of images, the content is as follows – Top left: Original RGB image; Top right: Dimension reduction of BrainSAIL embeddings for the image; Bottom: Two text queries using CLIP text branch showing language-indicated relevance results. **(c)** UMAP results on an inflated view of the brain for **S1**. **(d)** UMAP results on cortical flatmaps for **S2**, **S5** and **S7**. These results demonstrate that BrainSAIL can effectively localize semantically meaningful components of natural images and map them to appropriate brain regions. The cortical maps show color-coded mappings that align well with functionally-defined regions: body regions (EBA), face regions (FFA/aTL-faces), place regions (RSC/OPA/PPA), and food regions (yellow). Note that the food regions have been identified as flanking FFA by Jain et al. (2023), but we do not have independent functional localizer data for food for these subjects.

| Region | Faces | | | | Places | | | | Bodies | | | | Words | | | | Food | | | |
|---|---|---|---|---|---|---|---|---|---|---|---|---|---|---|---|---|---|---|---|---|
| | S1 | S2 | S5 | S7 | S1 | S2 | S5 | S7 | S1 | S2 | S5 | S7 | S1 | S2 | S5 | S7 | S1 | S2 | S5 | S7 |
| Face | **46** | **48** | **54** | **40** | 1 | 1 | 2 | 2 | 12 | 11 | 12 | 13 | 9 | 11 | 12 | 11 | 1 | 2 | 1 | 5 |
| Places | 1 | 1 | 1 | 3 | **76** | **80** | **89** | **75** | 0 | 2 | 3 | 2 | 9 | 12 | 11 | 9 | 8 | 7 | 10 | 7 |
| Bodies | 26 | 16 | 21 | 27 | 5 | 1 | 0 | 1 | **50** | **41** | **55** | **48** | 15 | 13 | 21 | **30** | 9 | 5 | 1 | 5 |
| Words | 1 | 7 | 3 | 8 | 6 | 3 | 2 | 9 | 8 | 6 | 7 | 7 | **38** | 28 | 26 | 23 | 16 | 13 | 17 | 35 |
| Food | 26 | 28 | 21 | 22 | 12 | 15 | 7 | 13 | 30 | 40 | 23 | 30 | 29 | **36** | **30** | 27 | **66** | **73** | **71** | **48** |

Table 2: **CLIP text alignment for each category selective brain region.** For each category selective brain region, we take the top-100 images from the NSD test set that elicit the highest fMRI response for each region. We then use BrainSAIL to compute the relevance maps for the top-100 images for each region. For each image, additional relevance maps are computed using the CLIP text encoder with text prompts from the five relevant categories. The text prompt with the highest Pearson correlation to the BrainSAIL relevance map is recorded as the category for that image. Units in %.

## 4 RESULTS

We utilize BrainSAIL to localize the semantic selectivity of different brain regions and demonstrate that the relevance maps are interpretable throughout the brain and correlate well with the known category-selective regions. We then explore the selectivity of higher visual cortex with respect to localized scene structure and image properties. Finally, we compare and contrast the localization results from three different vision foundation models. These results establish BrainSAIL as a novel technique for mapping and understanding the semantics of visual representations in the brain.

### 4.1 SETUP

We use the Natural Scenes Dataset (NSD; Allen et al. (2022)), the largest 7T fMRI dataset of human visual responses, focusing on four subjects (S1, S2, S5, S7) who viewed the full 10,000 image set (a subset of COCO images) three times each. fMRI activations (betas) were derived using GLMSingle (Prince et al., 2022) and normalized per session ($\mu = 0, \sigma^2 = 1$). Responses to repeated images were averaged. A brain encoder for each subject was trained on $\sim 9000$ unique images per subject, with the remaining $\sim 1000$ images viewed by all subjects being used for $R^2$ validation as the test set. Supplementary results for other subjects are included in the appendix. Face, place, body, and word regions were defined using independent category localizer data from NSD with a threshold of $t > 2$ (Stigliani et al., 2015). Food regions were defined using masks provided by Jain et al. (2023).

We train three encoders based on different neural network backbones. For all three, we utilize the ViT-Base model size. **(1)** For CLIP, we utilize OpenAI's official ViT-B/16 weights. This is a network trained on an infoNCE contrastive image-text objective. **(2)** For DINO, we utilize the latest official DINOv2 ViT-B/14+reg, and is a network trained on image-only self-supervision (Darcet et al., 2023). **(3)** For SigLIP, we utilize NVIDIA's implementation based on RADIOv2.5 ViT-B/16 (Ranzinger et al., 2024), as the original Google variant used a non-standard architecture. SigLIP utilizes a pairwise non-contrastive image-text objective (Zhai et al., 2023). All fMRI encoders are trained using MSE loss, with the backbone frozen. We validate the test time $R^2$ in Figure 6 and find that we achieve state-of-the-art results similar to Wang et al. (2023a) and Luo et al. (2024). We use CLIP for Sections 4.2 and 4.3, as it is the most widely used backbone in fMRI literature. We use 51 augmentation steps unless otherwise noted.

### 4.2 IMAGE FACTORIZATION USING THE BRAIN

To explore how different category-responsive regions in higher visual cortex align to different image parts we apply UMAP (McInnes et al., 2018) with an angular metric to linear brain weights and apply the same UMAP basis to dense features as produced by BrainSAIL. Note that during dimensionality reduction **we do not utilize any cortex category masks from NSD** – the region of interest outlines on the cortex in Figure 3a are for **visualization purposes only** and are derived from independent NSD functional localizers. As shown in Figure 3, we find that the factorization of the brain is well aligned to pre-identified functional regions, and broadly segments the cortex into axes along "people", "scenes" and "food". In particular, place regions, including the retrosplenial cortex (RSC), occipital place area (OPA), and parahippocampal place area (PPA), show selectivity for scene components (magenta). People regions, including the extrastriate body area (EBA), fusiform face area (FFA), occipital face area (OFA), show selectivity for face and body parts in the image (Green-Blue). Finally, we find that the recently identified food-responsive region that roughly surrounds FFA (Yellow) Khosla et al. (2022); Pennock et al. (2023); Jain et al. (2023) strongly corresponds to food in images. These results

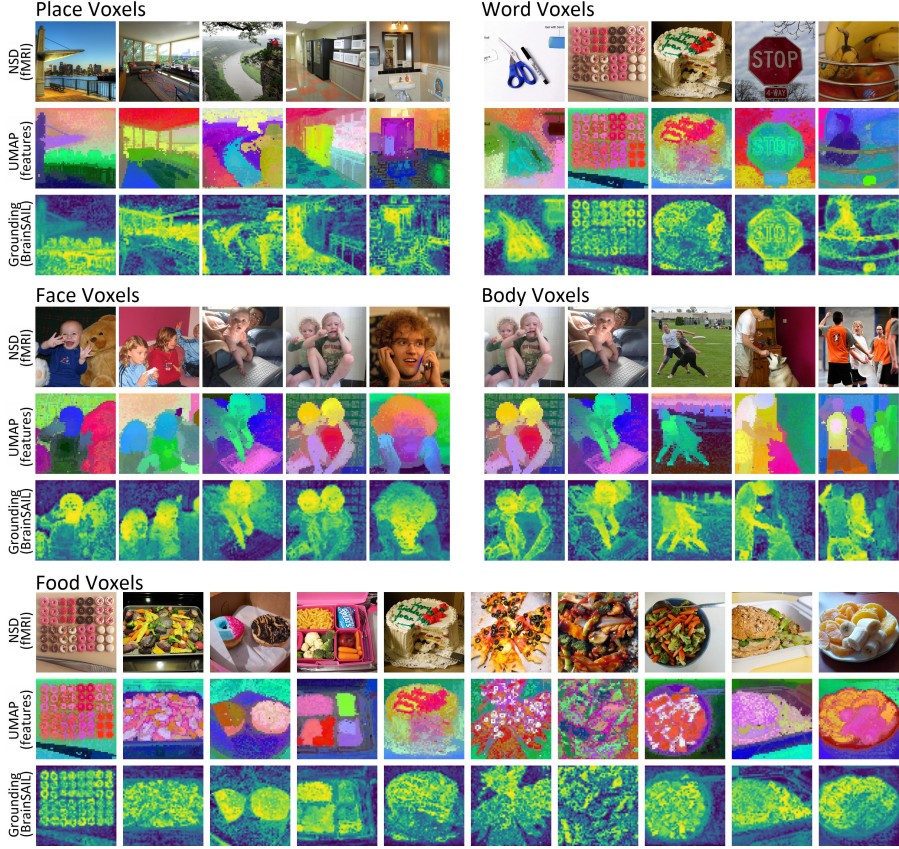

Figure 4: **Grounding results using BrainSAIL**. We visualize the top test set images as predicted by the CLIP fMRI encoder for each category selective region. For each image, we also visualize the image-wise UMAP for the distilled dense features. Note the UMAP basis here is computed imagewise, and not shared with Figure 3. The image-wise UMAP shows the semantic components present in the dense features. For each image, we further visualize the feature relevancy map for the category selective voxels illustrating that this method extracts the semantically relevant regions in complex compositional images.

establish that BrainSAIL can be used to characterize higher-level selectivity to individual semantic categories in complex natural images without prior knowledge of their semantic selectivity.

We further quantify the feature relevance maps for broad category selective regions in Figure 4 and Table 2. We use the brain encoder to predict the top-5 images for the place/word/face/body regions, and the top-10 images for the food-responsive region. We find that our method can effectively localize the objects relevant to each category- selective brain region. Note that the word region is known to have cross-selectivity to faces (Mei et al., 2010) and food (Khosla & Wehbe, 2022).

## 4.3 CORTEX SELECTIVITY TO IMAGE FEATURES

Going beyond semantic categories, we seek to explore the low- and mid-level image feature correlates that correspond to different brain regions. Prior work explored this by training a convolutional encoder on each NSD subject, which is limited to $\sim 10,000$ images each (Sarch et al., 2023). One concern is that using a small dataset with a convolutional backbone can lead to overfitting to the dataset's specific features and exacerbate the inherent biases of convolutional networks. To address this limitation, our method leverages vision transformers trained on massive datasets of hundreds of millions of images, thereby avoiding the hard-coded inductive biases present in CNNs (Raghu et al., 2021). We visualize BrainSAIL feature dissection results in Figure 5. Our method can successfully identify the known scene selective regions (RSC/OPA/PPA) as preferring high depth, and is successful even in OPA where Sarch et al. (2023) fails. We believe this is likely because OPA processes higher-level associative content and affordances (Bonner & Epstein, 2017; Aminoff & Tarr, 2021). Similarly, we identify the region surrounding FFA as being selective to high color saturation, which correspond to the food-responsive regions identified by Jain et al. (2023) and others. In OPA, we identify a split in color luminance preference, which is similar to the indoor/outdoor preferring regions identified

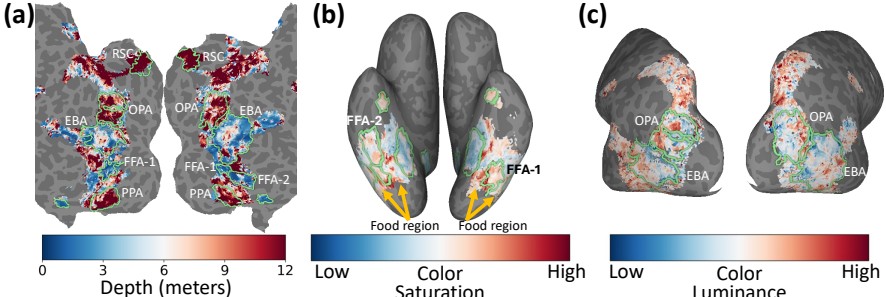

Figure 5: **Feature correlates with BrainSAIL**. We visualize the depth, color saturation, and color luminance (brightness) correlates for each brain region using BrainSAIL . **(a)** The scene selective regions, retrosplenial cortex (**RSC**), parahippocampal place area (**PPA**), and occipital place area (**OPA**) are all identified as having a preference for high depth. **(b)** On the ventral surface, we identify two stripes on each hemisphere, surrounding FFA with high saturation preference. These are the same brain regions identified by Jain et al. (2023) as being food selective. **(c)** In OPA, we identify an anterior/posterior split, where one region has high color luminance preference, and the other has low color luminance preference. This are the same regions identified by Luo et al. (2023) as being **outdoor/indoor** selective.

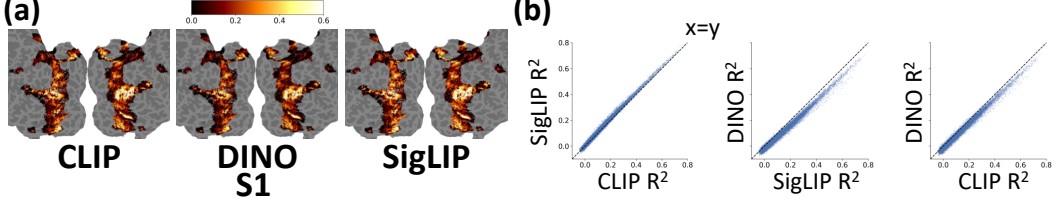

Figure 6: **Comparing the brain prediction performance for different encoder backbones**. **(a)** We validate each encoder $R^2$ on a test set and find that all three models achieve very high performance (comparable to Wang et al. (2023a)). **(b)** The voxel-wise correlation of test set $R^2$ for the three models. CLIP and SigLIP, which rely on language supervision, achieve higher performance than DINO (which trained via self-supervision with images).

by Lescroart & Gallant (2019), Peer et al. (2019), and Luo et al. (2023). These results demonstrate that our method can identify fine-grained selectivity with more broadly characterized brain regions.

## 4.4 ARE BRAIN ENCODERS EQUIVALENT?

Recent high-performing models such as CLIP, DINO, and SigLIP differ in their training objectives, architectures, and datasets: CLIP employs a contrastive image-language objective, DINO utilizes a self-supervised image loss without explicit linguistic guidance, and SigLIP leverages a non-contrastive pairwise image-language loss. Despite these differences, when employed as the backbone for fMRI encoders, these models exhibit similar performance in predicting brain responses, achieving comparable $R^2$ values on the test set as shown in Figure 6. This observation raises an important question about the nature of each model's learned features and their alignment with one another: Do these models converge upon similar feature representations (Chen & Bonner, 2024) for category selective brain regions despite their varied training paradigms?

To investigate the representational differences between the models, we perform BrainSAIL analysis for scene-, face-, and food-responsive brain regions and qualitatively visualize the results in Figure 8. While all three models exhibit broad similarities in their grounding maps, DINO, trained without language supervision, demonstrates a stronger sensitivity to low-level visual features compared to CLIP and SigLIP (Wang et al., 2023a). This is evident in the food-responsive region (Figure 8), where DINO's grounding map for a pizza image excludes the toppings and misses differently colored vegetables on a metal plate, suggesting a focus on color and texture rather than the concept of "food" itself. Similarly, in the face region, DINO's grounding map exhibits less reliance on semantically relevant features such as eyes, nose, and mouth. We hypothesize that this greater sensitivity to visual features in DINO stems from its lack of language guidance during training, preventing it from learning the higher-level semantic correlations that link visually disparate parts and objects within a category. As high-performing "proxy models" of visual brain representation (Leeds et al., 2013), these and other underlying model characteristics – architecture, training objective, training dataset,

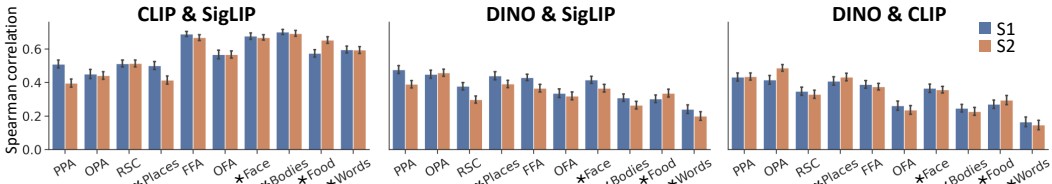

Figure 7: **Model similarity across ROIs**. Brain encoder backbone spatial similarity for the ground truth top-100 images from the test set for each category-selective brain region. A ⋆ denotes a domain-defined network of regions encompassing multiple ROIs. CLIP and SigLIP relevance maps are more similar to one another than either is to DINO. Error bars indicate standard error across the 100 images.

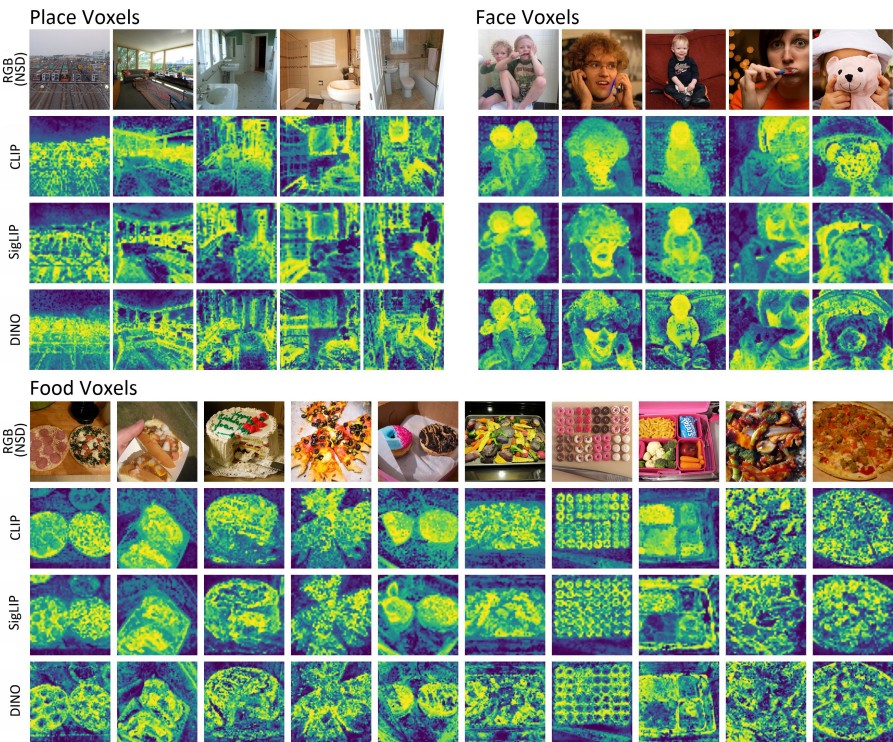

Figure 8: **Comparing different brain encoder backbones with BrainSAIL**. Visualization of the top test set images for the place, face, and food category-selective brain regions as predicted by CLIP, SigLIP, and DINO. While all models show broadly similar feature relevance for a given brain area, there are important differences. DINO, with no language supervision, exhibits greater sensitivity to visual similarity, at the cost of semantic coherence.

etc. – are important considerations for developing more robust encoding models that can bridge the gap between artificial and biological vision systems.

## 5 DISCUSSION

**Limitations and Future Work.**  BrainSAIL achieves strong localization performance and benefits from a pre-trained vision transformer, reducing reliance on the fMRI dataset for backbone training. However, it is still necessary to train the fMRI encoder on these data, and thus potential dataset biases in the human neural data and how it was collected can influence the learned representations and conclusions (Shirakawa et al., 2024). Future work should explore training on larger and more diverse neural datasets to mitigate this limitation and enhance the generalizability of our findings.

**Conclusion.**  We propose BrainSAIL, a method that leverages vision foundation models to interrogate which semantic components of complex natural images lead to the neural activation of specific regions of the brain. Based on the vision transformer architecture, we: (1) semantically attribute and localize relevant objects in complex compositional images; (2) jointly factorize images and semantically selective regions in the human brain; (3) identify the feature correlates of depth, saturation, and luminance that underlie semantic selectivity; (4) explicate differences in fMRI encoders that achieve similar overall brain prediction performance. *In toto,* these results establish that BrainSAIL is a powerful new approach to data-driven explorations of the human higher visual cortex.

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

# A  APPENDIX

## Sections

### A.1 Implementation and fMRI Processing Details

**Encoder training.** Our experiments utilize a mixture of GPUs including Nvidia V100 (16GB and 32GB), 2080 Ti, A6000, 6000Ada, L40S, and 4090 cards. The network training code is implemented in PyTorch. For encoder training, we employ the Adam optimizer with a decoupled weight decay of $2 \times 10^{-2}$. The initial learning rate is set to $3 \times 10^{-4}$ and decays exponentially to $1.5 \times 10^{-4}$ over 100 epochs. Each subject is trained independently. All backbone networks are frozen, and operate in fp16 mode.

For encoder training, we always utilize the network's native input resolution. For CLIP, we resize images to $224 \times 224$. For DINO, we resize images to $518 \times 518$. For SigLIP using the AM-RADIO backbone, and we resize images to $768 \times 768$. After resizing, we augment the images with random pixel-wise value scaling between 0.95 and 1.05, followed by normalization using each network's respective image mean and variance. Before network input, images are randomly offset by up to 4 pixels horizontally and vertically, with edge padding filling the resulting empty pixels. Independent Gaussian noise ($\mu = 0$, $\sigma^2 = 0.05$) is added to each pixel.

**Feature extraction.** We perform positional encoding interpolation for CLIP and DINO networks in order to extract higher resolution embeddings. For `CLIP ViT-B/16` we modify the network to accept images of $896 \times 896$ (4x upsampling; $56 \times 56$ final patch resolution); For `DINOv2 ViT-B/14+reg` we modify the network to accept images of $1036 \times 1036$ (2x upsampling; $74 \times 74$ final patch resolution); For SigLIP based on Nvidia's `RADIOv2.5 ViT-B/16`, we do not perform upsampling, and have $48 \times 48$ final patch resolution. As noted in the AM-RADIO paper, the spatial patch features of these networks are highly robust to positional encoding upsampling. We perform 51 augmentation steps for CLIP, where the first augmentation step is null (no shift or flipping). For DINO and SigLIP, as these networks contain registers which mitigate the artifacts to a certain degree, we perform 25 augmentation steps.

For CLIP, we utilize the NACLIP self-attention modification, with gaussian std set to 10.0. For DINO and SigLIP, we use the MaskCLIP self-attention modification. For SigLIP specifically, since we are using the AM-RADIO implementation, we apply their provided SigLIP adapter head to the extracted features. As shown in section A.7, this enables zero-shot probing with the official SigLIP text encoder using the extracted dense features.

**Computational cost.** Our "Learning-Free Distillation Module" is highly efficient. With pre-extracted dense features for each augmentation, we observe our procedure generally takes less than 0.2 seconds on a GPU, this is roughly $100\times$ faster than "Denoising Vision Transformers" on the same hardware and similarly ignoring the feature extraction cost.

**CLIP sentences.** We define a set of natural language captions to help us evaluate the alignment between fMRI region-wise relevance maps and concepts. For every caption and every image, we compute a relevance map using the CLIP text encoder. The category that contains the caption with the highest pearson correlation to the fMRI relevance map is assigned as the category.

```
face_class = ["A face facing the camera", "A photo of a face", "A
photo of a human face", "A photo of faces", "A photo of a person's
face", "A person looking at the camera", "People looking at the
camera","A portrait of a person", "A portrait photo"]

body_class = ["A photo of a torso", "A photo of torsos", "A photo
of limbs", "A photo of bodies", "A photo of a person", "A photo
of people", "A photo of a body", "A person's arms", "A person's
legs", "A photo of hands"]

scene_class = ["A photo of a bedroom", "A photo of an office","A
photo of a hallway", "A photo of a doorway", "A photo of interior
design", "A photo of a building", "A photo of a house", "A photo
of nature", "A photo of landscape", "A landscape photo", "A photo
of trees", "A photo of grass"]

food_class = ["A photo of food", "A photo of cuisine", "A photo
of fruit", "A photo of foodstuffs", "A photo of a meal", "A photo
of bread", "A photo of rice", "A photo of a snack", "A photo of
```

```
pastries", "A photo of vegetables", "A photo of pizza", "A photo
of soup", "A photo of meat", "A photo of candy"]

text_class = ["A photo of words", "A photo of glyphs", "A photo of
a glyph", "A photo of text", "A photo of numbers", "A photo of a
letter", "A photo of letters", "A photo of writing", "A photo of
text on an object"]
```

**fMRI data processing.** All of our analysis is done in subject native space voxels in $1.8mm^3$ resolution (`func1pt8mm`), and we use the beta values from `betas_fithrf_GLMdenoise_RR`. Cortical voxels are selected based on the nsdgeneral mask.

In order to select visually responsive voxels for Figure 3, we utilize the HCP parcellation which yields 180 regions. For each region in each subject, we compute the average noise ceiling using NSD provided data. We rank the regions within each subject, and average the ranks across all subjects. The best 25% (45 out of 180) regions by noise ceiling are selected. We further mask out voxels which are labeled as early visual (V1~ V4).

**Intuition for learning-free distillation.** We are specifically motivated by literature in learning dense semantic embeddings. In particular, Kobayashi et al. (2022) proposed learning a dense 3D semantic representation. However Kobayashi et al. (2022) uses LSeg (Li et al., 2022) – a supervised open-vocabulary segmentation network which demonstrates relatively poor generalization to categories outside of the training set (Kerr et al., 2023). As we do not wish to restrict the hypothesis space relative to cortical selectivity, we were motivated to explore methods which preserve the original representational capability of vision foundation models. By default, the dense token embeddings do not lie in the same semantic space as the summary token. However modifications like MaskCLIP (Zhou et al., 2022), SCLIP (Wang et al., 2023b), and NACLIP (Hajimiri et al., 2024) propose changes to the self-attention mechanism that ensure the dense embeddings are in the same space as the summary token. We utilize NACLIP for CLIP (or models without registers). As register tokens have not been explored in the context of self-attention modifications, we use MaskCLIP for models with registers. Without further distillation, these methods yield dense embeddings that have artifacts.

Recent work has proposed to learn consistent 3D representations using NeRF and feature fields (Mildenhall et al., 2021; Kobayashi et al., 2022). These methods demonstrate that a spatially consistent representation can be learned in the presence of noise (pixel noise or measurement noise) by utilizing a photometric 3D consistency framework with multiple views. As we do not have multiple views of a single image in 2D, we augment images with horizontal flips and vertical/horizontal offsets to generate synthetic views. These augmentations do not change the semantics of a scene. We observe that because the artifacts are not static relative to scene content, augmentations can effectively render artifacts into "measurement noise".

Originally NeRF and feature fields perform very expensive neural field learning with gradients and MSE loss. Faster learning and inference procedures are required to facilitate data-driven studies of the visual cortex using thousands of images. We show that averaging over views achieves the same mathematical result as MSE driven optimization.

## A.2 COMPARING CLIP GRANULARITY AND BRAIN SALIENCY

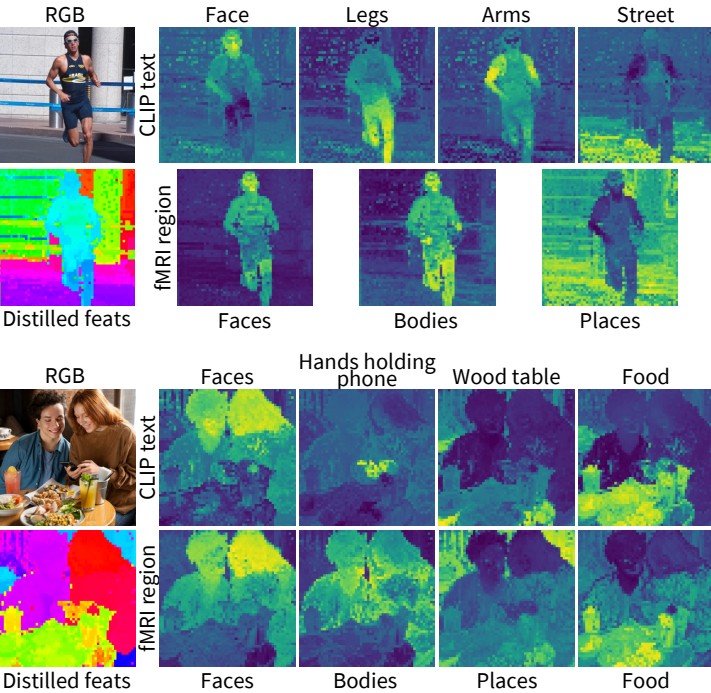

Figure S.1: **Comparing the granularity of CLIP embeddings with brain region saliency**. On the left, we visualize the image and the UMAP of the distilled NACLIP embeddings. On the top right, we probe the dense features using the CLIP-text encoder. On the bottom right, we visualize the relevance map using the fMRI encoder weights for $t > 2$ functional regions from **S1**. In both cases, the probing is done with a cosine similarity. We find that brain data usually has semantically coarser attributions. Likely due to NSD data co-occurrence statistics, we find that neural data does not consistently separate faces & bodies, while CLIP itself can have very fine-grained representations.

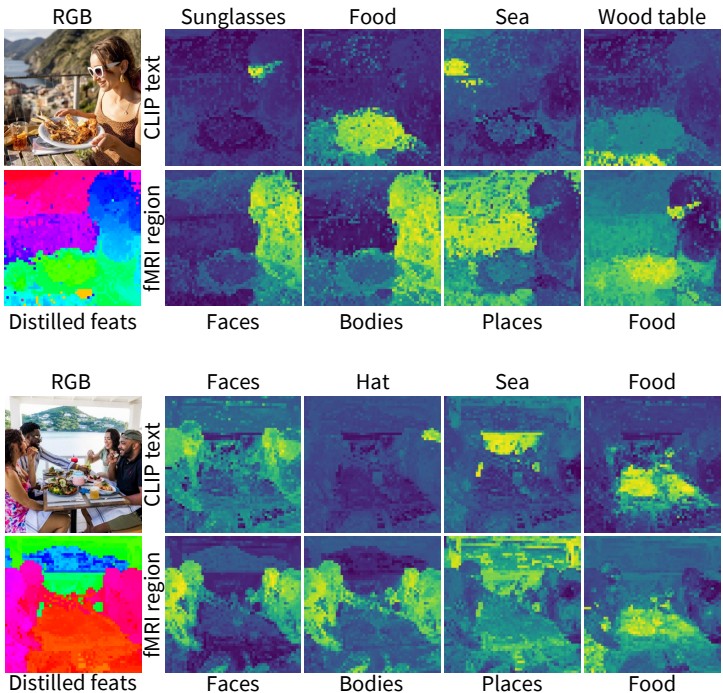

Figure S.2: **Comparing the granularity of CLIP embeddings with brain region saliency**. On the left, we visualize the image and the UMAP of the distilled NACLIP embeddings. On the top right, we probe the dense features using the CLIP-text encoder. On the bottom right, we visualize the relevance map using the fMRI encoder weights for $t > 2$ functional regions from **S1**. In both cases, the probing is done with a cosine similarity. We find that brain data usually has semantically coarser attributions. Likely due to NSD data co-occurrence statistics, we find that neural data does not consistently separate faces & bodies, and attributes food selectivity to small objects or high saturation image regions (note sunglasses in top image, and pink dress of bottom image) – food itself is often both small and high saturation. Our method can reveal surprising behavior of fMRI encoders.

## A.3 ADDITIONAL ENCODER TEST SET $R^2$ FOR CLIP, DINO, AND SIGLIP

Subjects 1, 2, 5, 7 are those that completed the full scan of $10,000$ images – where each image was viewed 3 times. They are also the subjects reported to have the highest noise ceiling in NSD (higher is better).

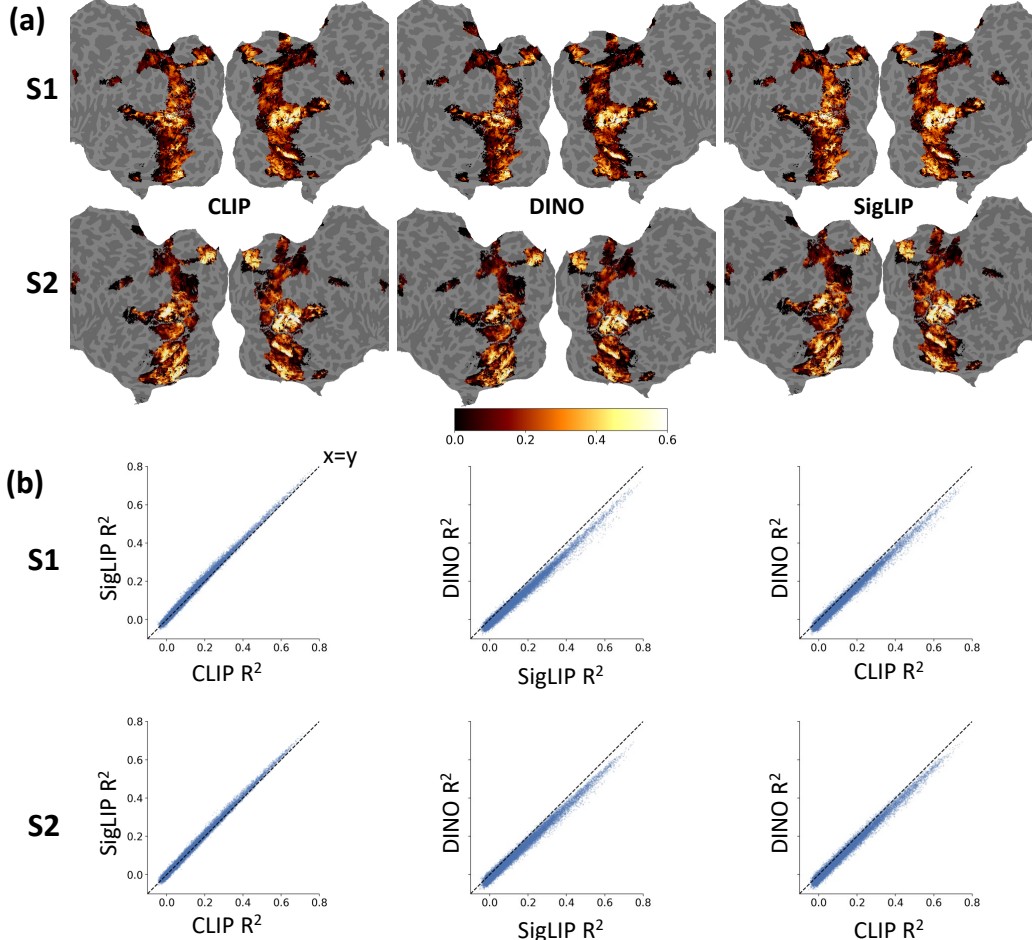

Figure S.3: **Test set $R^2$ using different encoder backbones on subjects S1 & S2.** (a) We validate each encoder $R^2$ on a test set and find that all three models achieve very high performance (comparable to Wang et al. (2023a)). (b) The voxel-wise correlation of test set $R^2$ for the three models. CLIP and SigLIP, which rely on language supervision, achieve higher performance than DINO (which trained via self-supervision with images).

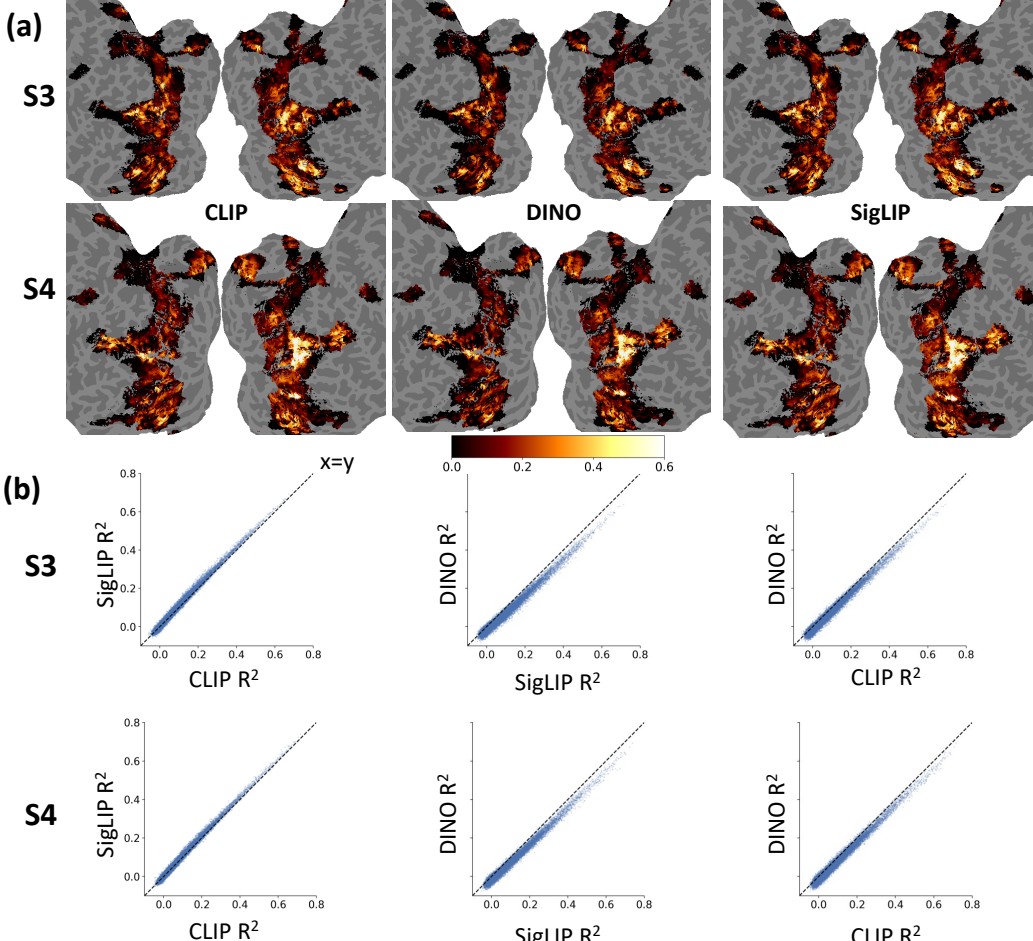

Figure S.4: **Test set $R^2$ using different encoder backbones on subjects S3 & S4. (a)** We validate each encoder $R^2$ on a test set and find that all three models achieve very high performance (comparable to Wang et al. (2023a)). **(b)** The voxel-wise correlation of test set $R^2$ for the three models. CLIP and SigLIP, which rely on language supervision, achieve higher performance than DINO (which trained via self-supervision with images).

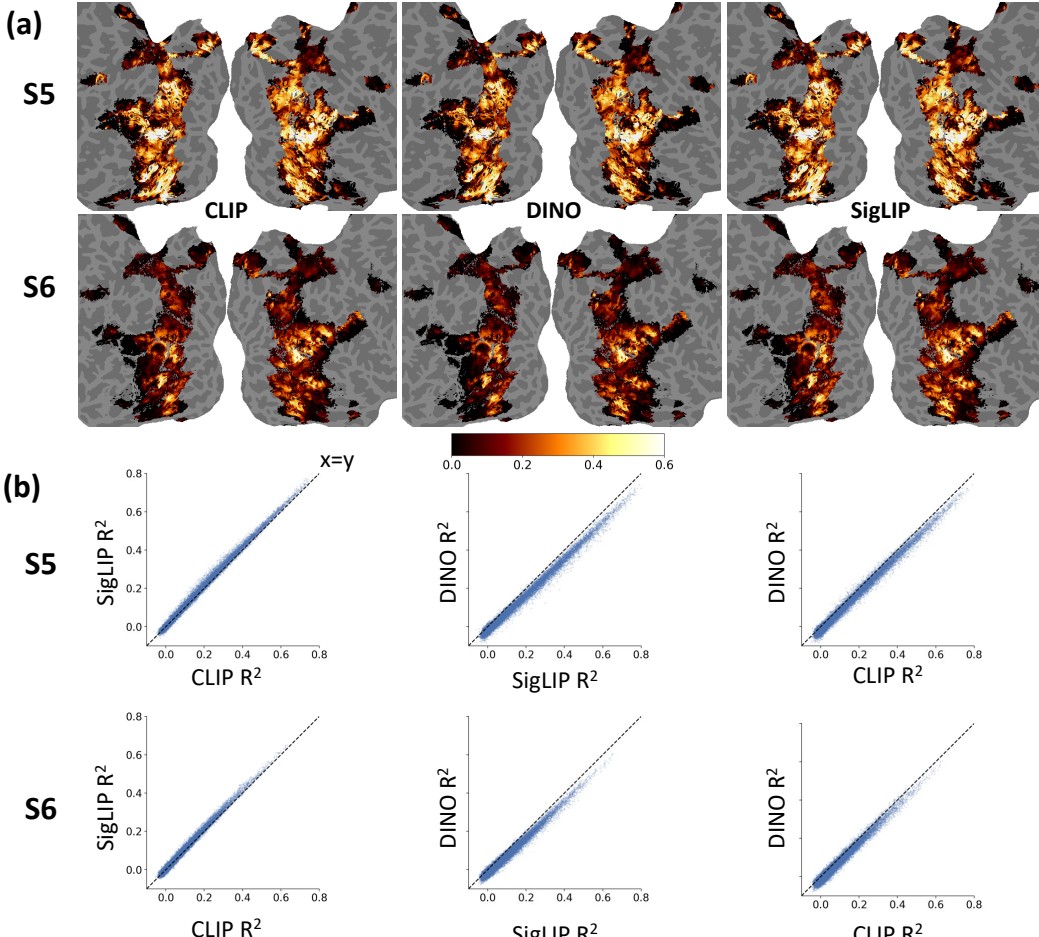

Figure S.5: **Test set $R^2$ using different encoder backbones on subjects S5 & S6.** **(a)** We validate each encoder $R^2$ on a test set and find that all three models achieve very high performance (comparable to Wang et al. (2023a)). **(b)** The voxel-wise correlation of test set $R^2$ for the three models. CLIP and SigLIP, which rely on language supervision, achieve higher performance than DINO (which trained via self-supervision with images).

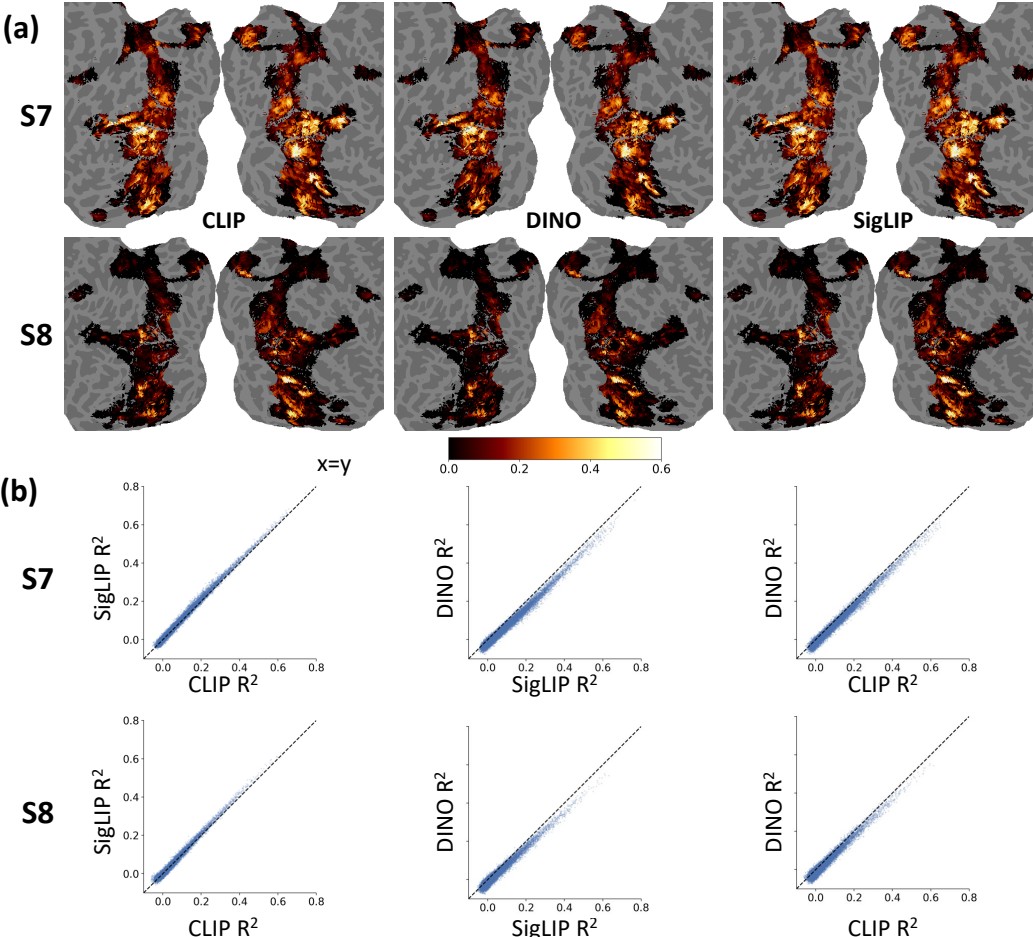

Figure S.6: **Test set $R^2$ using different encoder backbones on subjects S7 & S8.** (a) We validate each encoder $R^2$ on a test set and find that all three models achieve very high performance (comparable to Wang et al. (2023a)). (b) The voxel-wise correlation of test set $R^2$ for the three models. CLIP and SigLIP, which rely on language supervision, achieve higher performance than DINO (which trained via self-supervision with images).

## A.4 VISUALIZATION OF GROUND TRUTH FUNCTIONAL LOCALIZER STATISTICS

In this section, we show the ground truth functional localizer t-statistic provided by NSD. As NSD does not provide a food localizer, we obtain the food mask from Jain et al. (2023). During the UMAP, we do not provide our method with the ground truth functional localizer information, and these t-statistics are provided here for reference only.

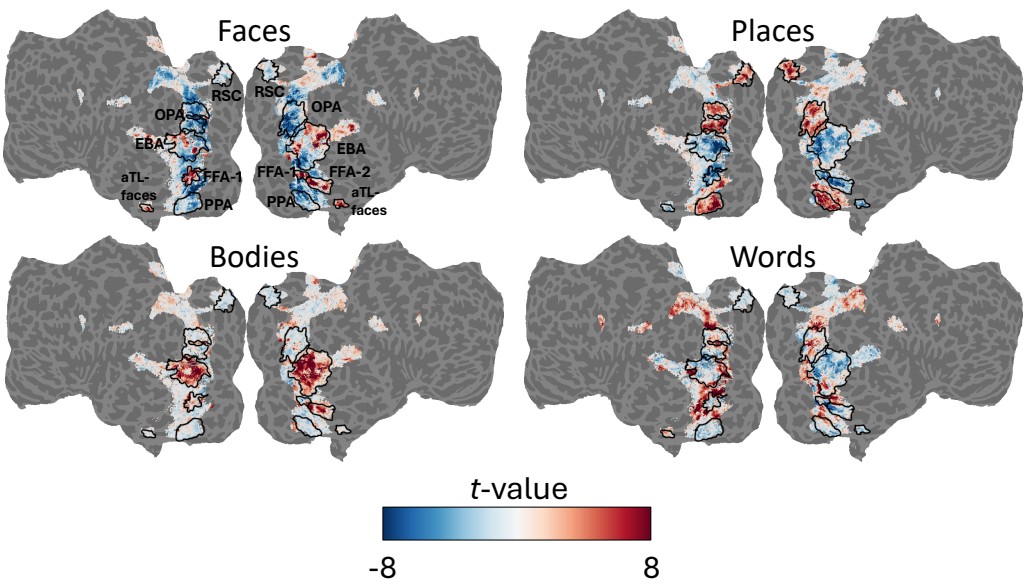

Figure S.7: **Semantic category functional localizer statistics for S1.** We visualize the ground truth functional localizer t-statistic provided by NSD. Higher indicates stronger selectivity to a semantic category.

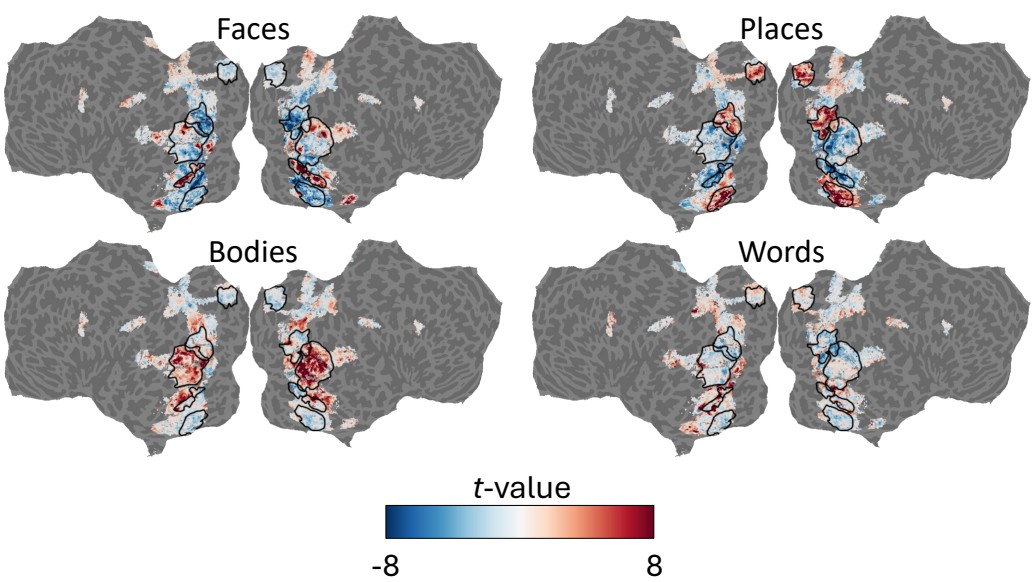

Figure S.8: **Semantic category functional localizer statistics for S2.** We visualize the ground truth functional localizer t-statistic provided by NSD. Higher indicates stronger selectivity to a semantic category.

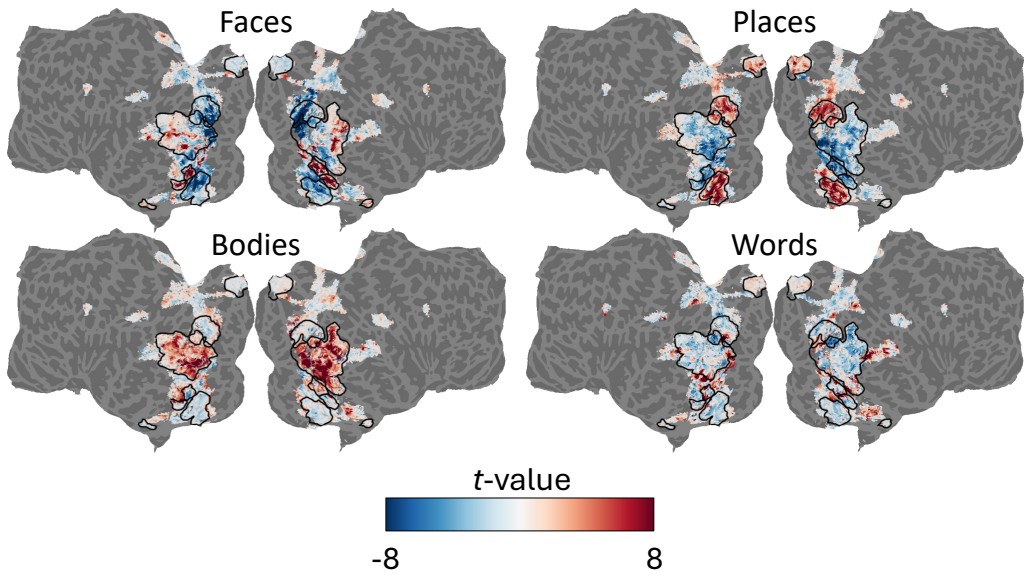

Figure S.9: **Semantic category functional localizer statistics for S5.** We visualize the ground truth functional localizer *t*-statistic provided by NSD. Higher indicates stronger selectivity to a semantic category.

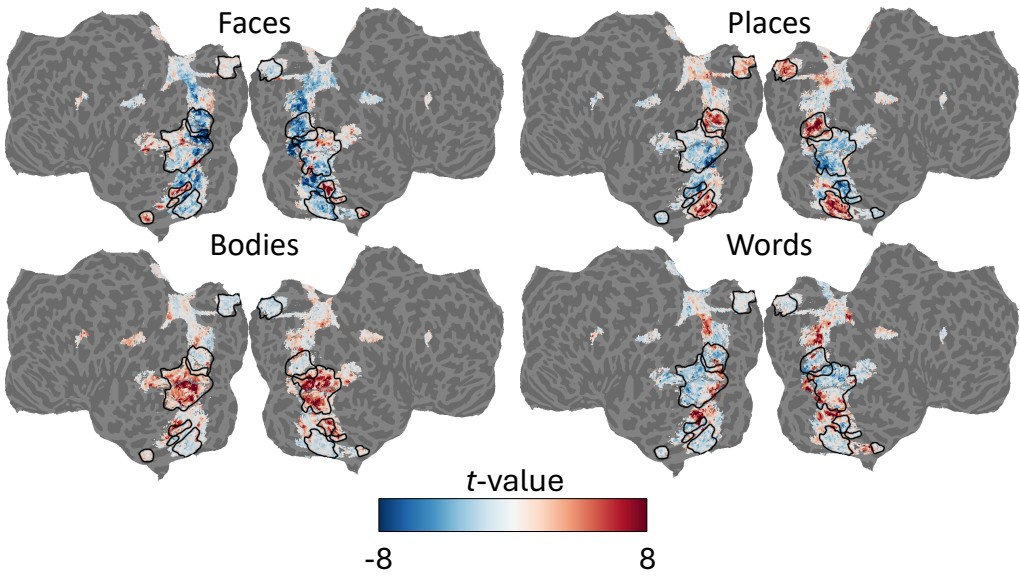

Figure S.10: **Semantic category functional localizer statistics for S7.** We visualize the ground truth functional localizer *t*-statistic provided by NSD. Higher indicates stronger selectivity to a semantic category.

### A.5 VISUALIZATION OF ENCODER WEIGHT UMAP FOR ALL SUBJECTS USING CLIP BACKBONE

We utilize OpenAI's official `ViT-B/16` as the encoder backbone, and visualize the UMAP as applied to the fMRI encoder weights and dense features.

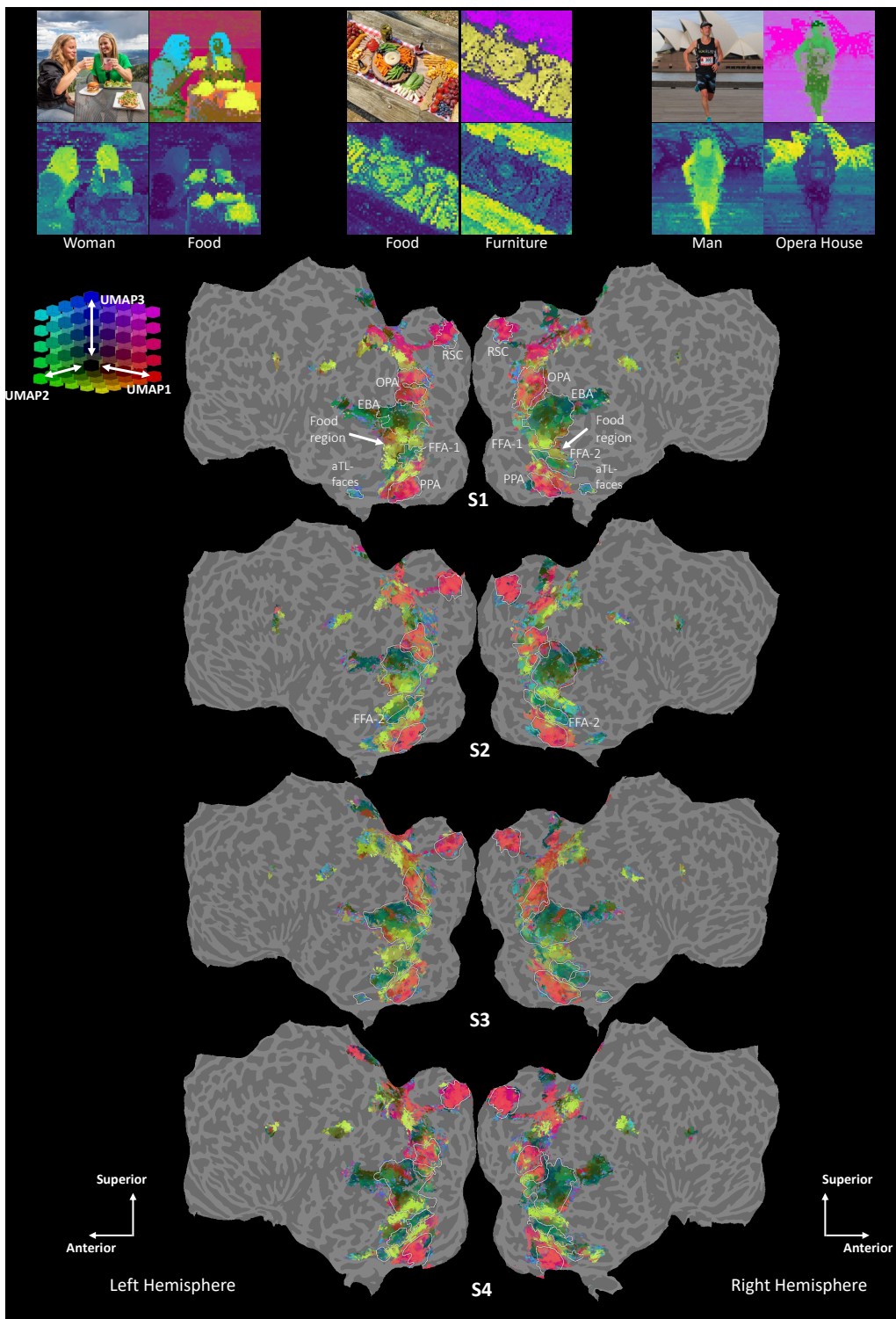

Figure S.11: **Encoder with CLIP backbone UMAP transform results for subjects S1-S4.** We normalize all encoder weights to unit norm prior to UMAP, and use an angular metric for the fitting and projection stage. The UMAP is fitted on S1 and reused across all subjects and images. Both patch and voxel vectors are projected onto the space of natural images prior to transform using softmax weighted sum similar to BrainSCUBA. For each quartet of images, the content is as follows – Top left: Original RGB image; Top right: Dimension reduction of BrainSAIL embeddings for the image using CLIP features; Bottom: Two text queries using CLIP text branch showing language-indicated relevance results.

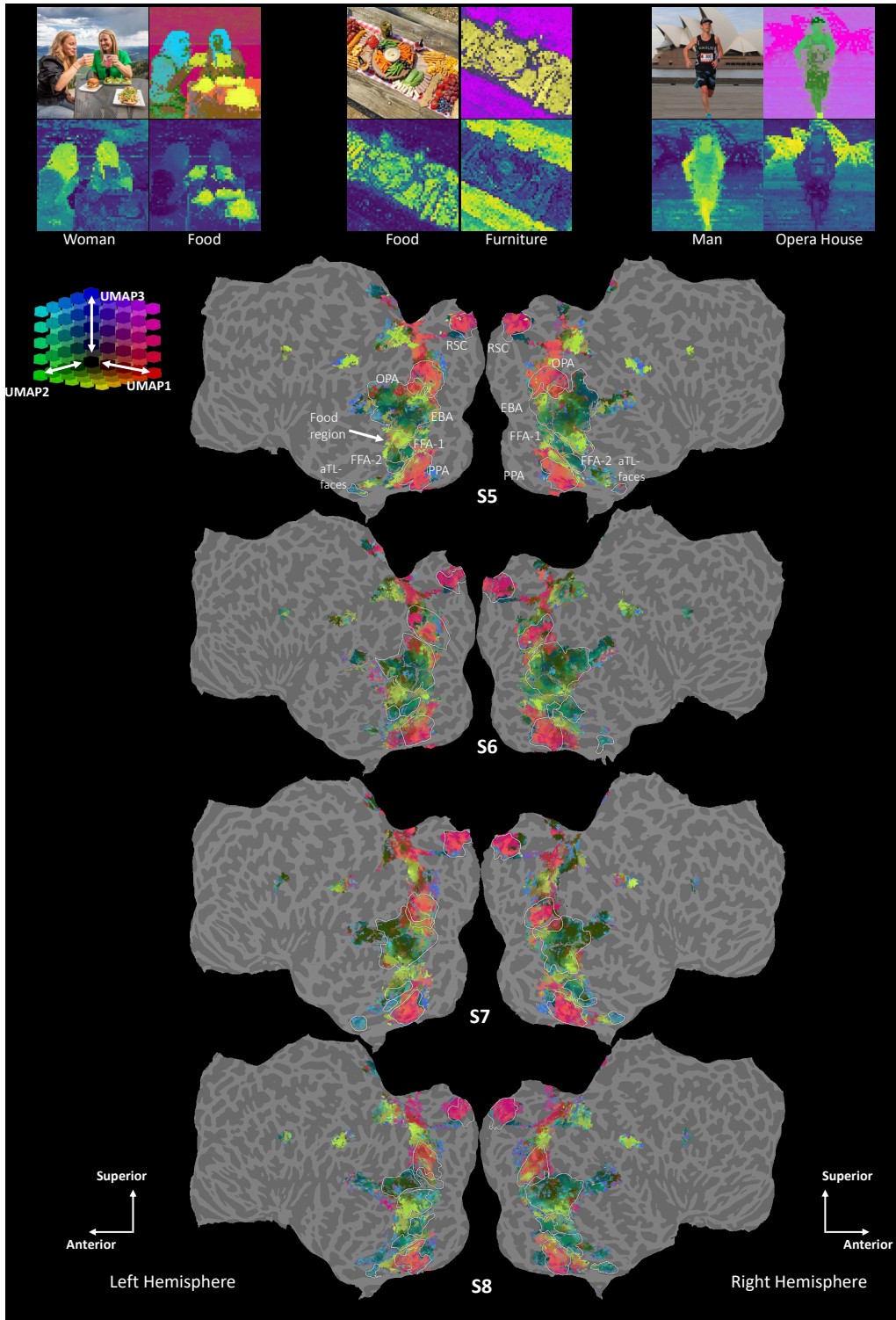

Figure S.12: **Encoder with CLIP backbone UMAP transform results for subjects S5-S8.** We normalize all encoder weights to unit norm prior to UMAP, and use an angular metric for the fitting and projection stage. The UMAP is fitted on S1 and reused across all subjects and images. Both patch and voxel vectors are projected onto the space of natural images prior to transform using softmax weighted sum similar to BrainSCUBA. For each quartet of images, the content is as follows – Top left: Original RGB image; Top right: Dimension reduction of BrainSAIL embeddings for the image using CLIP features; Bottom: Two text queries using CLIP text branch showing language-indicated relevance results.

## A.6 Visualization of encoder weight UMAP for all subjects using DINO backbone

We utilize Meta's official `DINOv2 ViT-B/14+reg` as the encoder backbone, and visualize the UMAP as applied to the fMRI encoder weights and dense features. The UMAP method is nonparametric , and the output can depend on the random seed. Similar colors for UMAP applied to DINO encoder weights does not indicate any specific similarity to CLIP results in the prior section.

As DINO does not have a text encoder, we cannot visualize text-based localization probes.

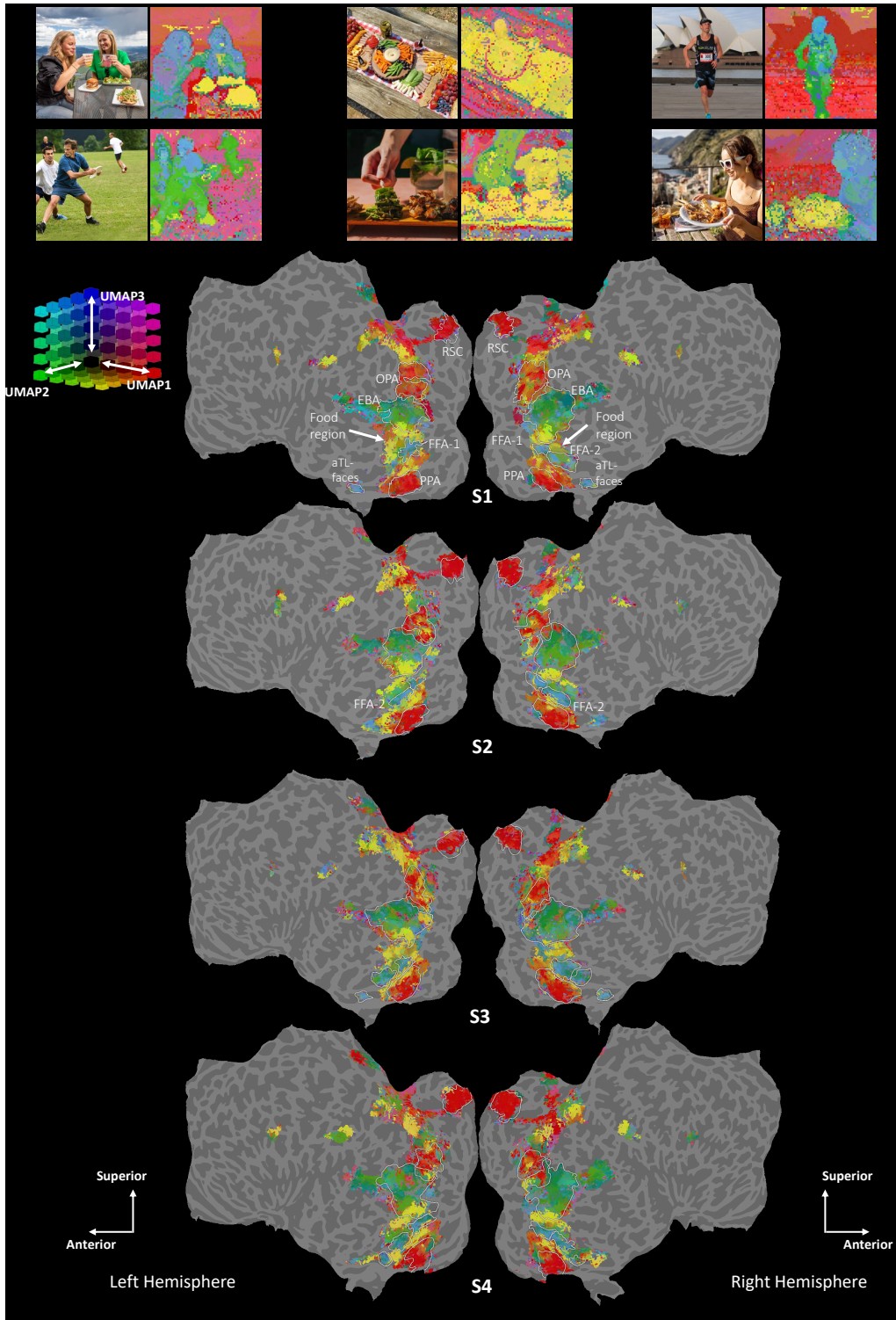

Figure S.13: **Encoder with DINO backbone UMAP transform results for subjects S1-S4.** We normalize all encoder weights to unit norm prior to UMAP, and use an angular metric for the fitting and projection stage. The UMAP is fitted on S1 and reused across all subjects and images. Both patch and voxel vectors are projected onto the space of natural images prior to transform using softmax weighted sum similar to BrainSCUBA. For each pair of images, the content is as follows – Left: Original RGB image; Right: Dimension reduction of BrainSAIL embeddings for the image using DINO features.

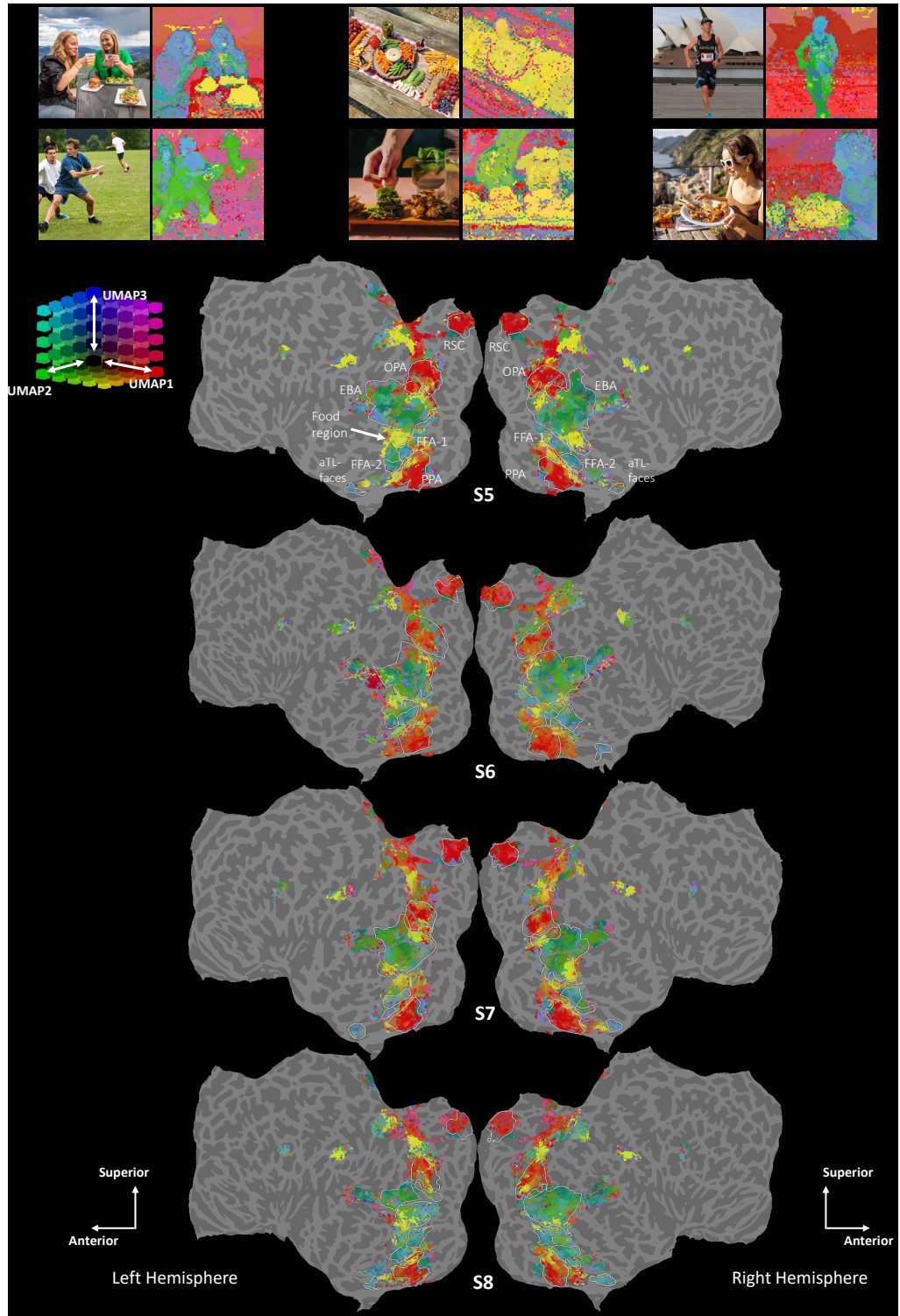

Figure S.14: **Encoder with DINO backbone UMAP transform results for subjects S5-S8.** We normalize all encoder weights to unit norm prior to UMAP, and use an angular metric for the fitting and projection stage. The UMAP is fitted on S1 and reused across all subjects and images. Both patch and voxel vectors are projected onto the space of natural images prior to transform using softmax weighted sum similar to BrainSCUBA. For each pair of images, the content is as follows – Left: Original RGB image; Right: Dimension reduction of BrainSAIL embeddings for the image using DINO features.

## A.7 Visualization of encoder weight UMAP for all subjects using SigLIP backbone

We utilize Nvidia's `RADIOv2.5 ViT-B/16` implementation for SigLIP. Nvidia's implementation was choosen as the original Google implementation did not utilize a ViT with a attention `[CLS]` token. We visualize the UMAP as applied to the fMRI encoder weights and dense features. The UMAP method is nonparametric , and the output can depend on the random seed. Similar colors for UMAP applied to SigLIP encoder weights does not indicate any specific similarity to DINO or CLIP results in the prior section.

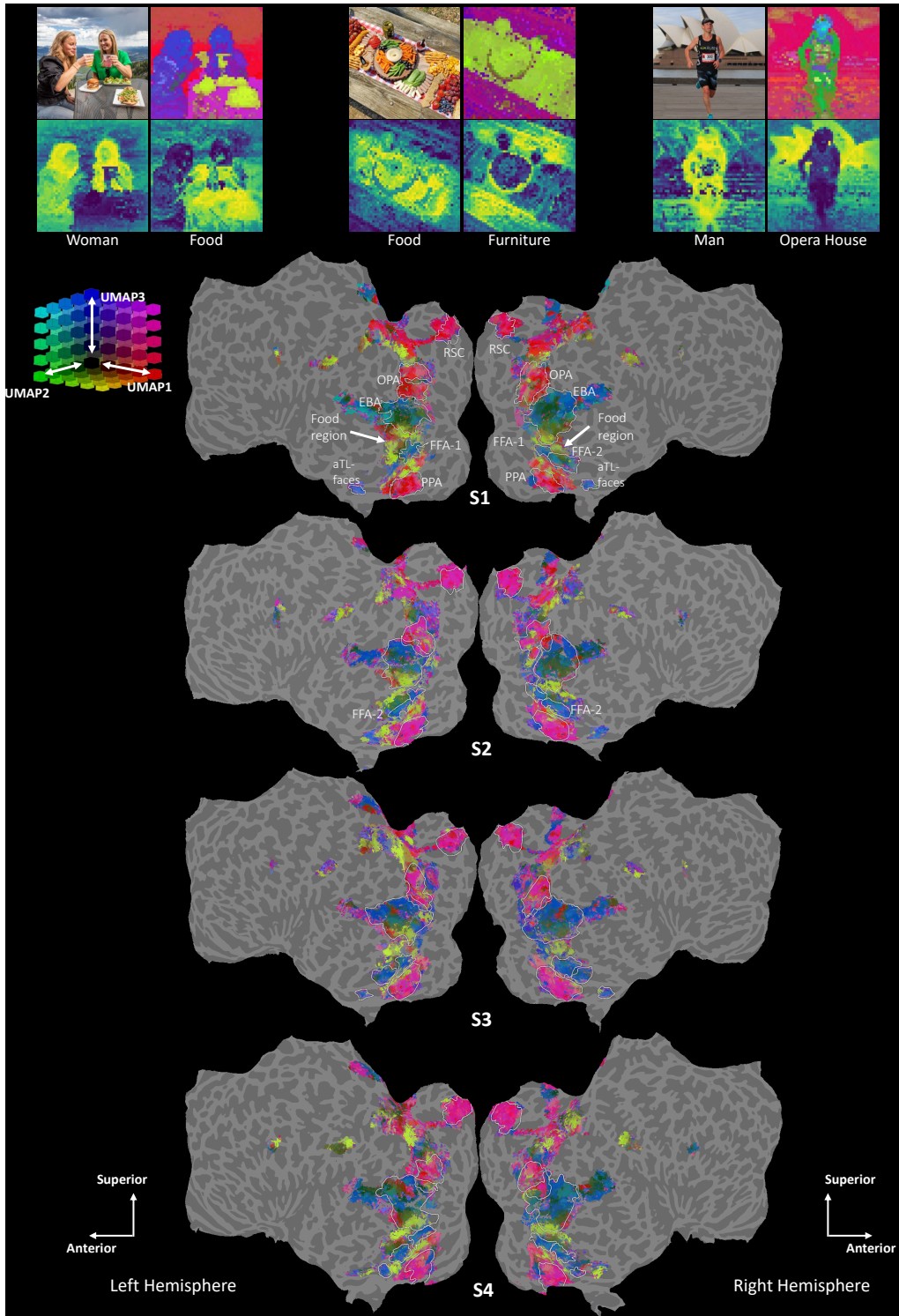

Figure S.15: **Encoder with SigLIP backbone UMAP transform results for subjects S1-S4.** We normalize all encoder weights to unit norm prior to UMAP, and use an angular metric for the fitting and projection stage. The UMAP is fitted on S1 and reused across all subjects and images. Both patch and voxel vectors are projected onto the space of natural images prior to transform using softmax weighted sum similar to BrainSCUBA. For each quartet of images, the content is as follows – Top left: Original RGB image; Top right: Dimension reduction of BrainSAIL embeddings for the image using SigLIP features; Bottom: Two text queries using SigLIP text branch showing language-indicated relevance results.

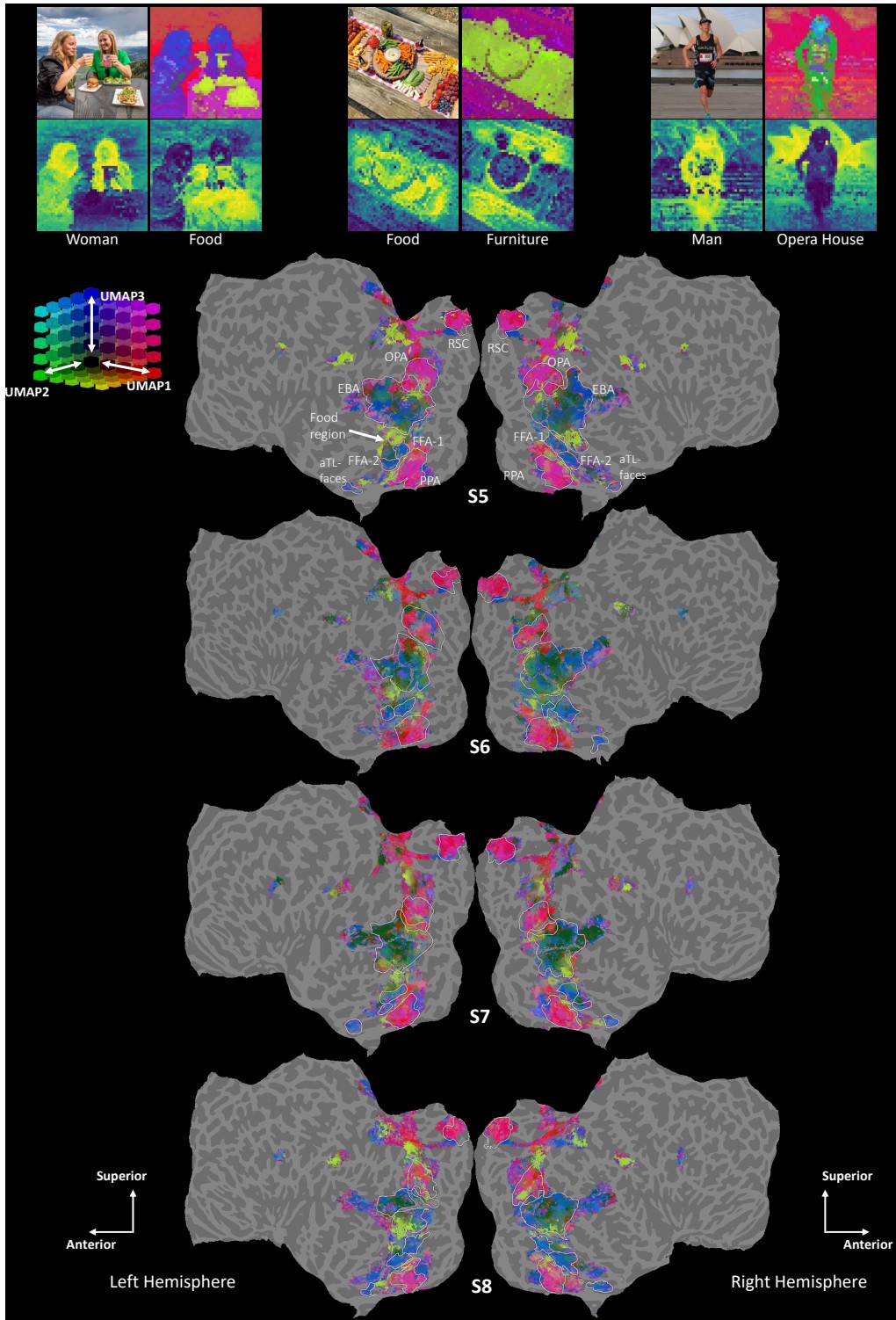

Figure S.16: **Encoder with SigLIP backbone UMAP transform results for subjects S5-S8.** We normalize all encoder weights to unit norm prior to UMAP, and use an angular metric for the fitting and projection stage. The UMAP is fitted on S1 and reused across all subjects and images. Both patch and voxel vectors are projected onto the space of natural images prior to transform using softmax weighted sum similar to BrainSCUBA. For each quartet of images, the content is as follows – Top left: Original RGB image; Top right: Dimension reduction of BrainSAIL embeddings for the image using SigLIP features; Bottom: Two text queries using SigLIP text branch showing language-indicated relevance results.

## A.8 ADDITIONAL VISUALIZATION OF GROUNDING USING CLIP BACKBONE

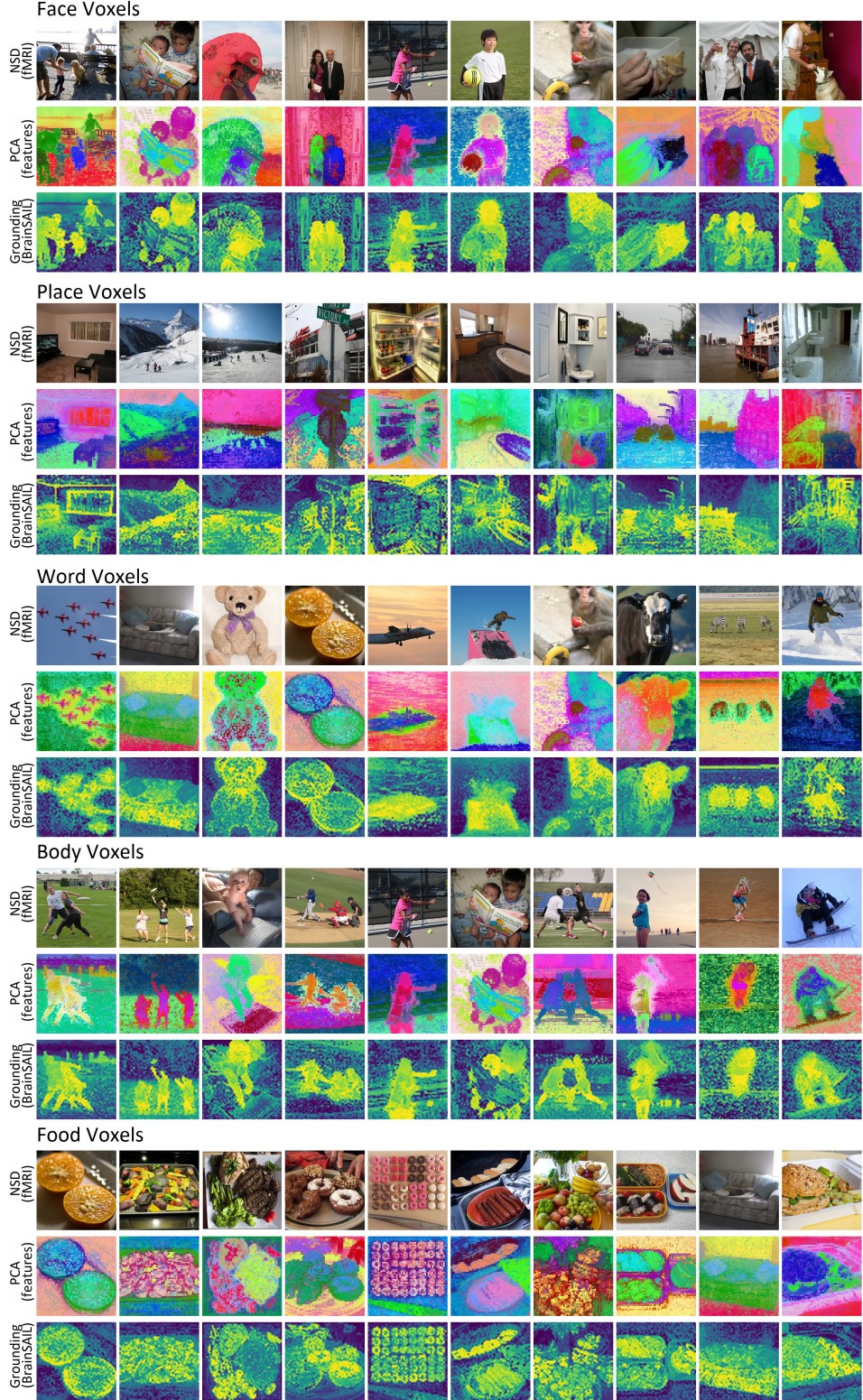

Figure S.17: **Grounding results using BrainSAIL for S1 CLIP backbone.**. We visualize the top test set images according to NSD fMRI for each category selective region. For each image, we also visualize the image-wise PCA for the distilled dense features. Note the PCA basis here is computed imagewise. For each image, we further visualize the feature relevancy map for the category selective voxels illustrating that this method extracts the semantically relevant regions in complex compositional images.

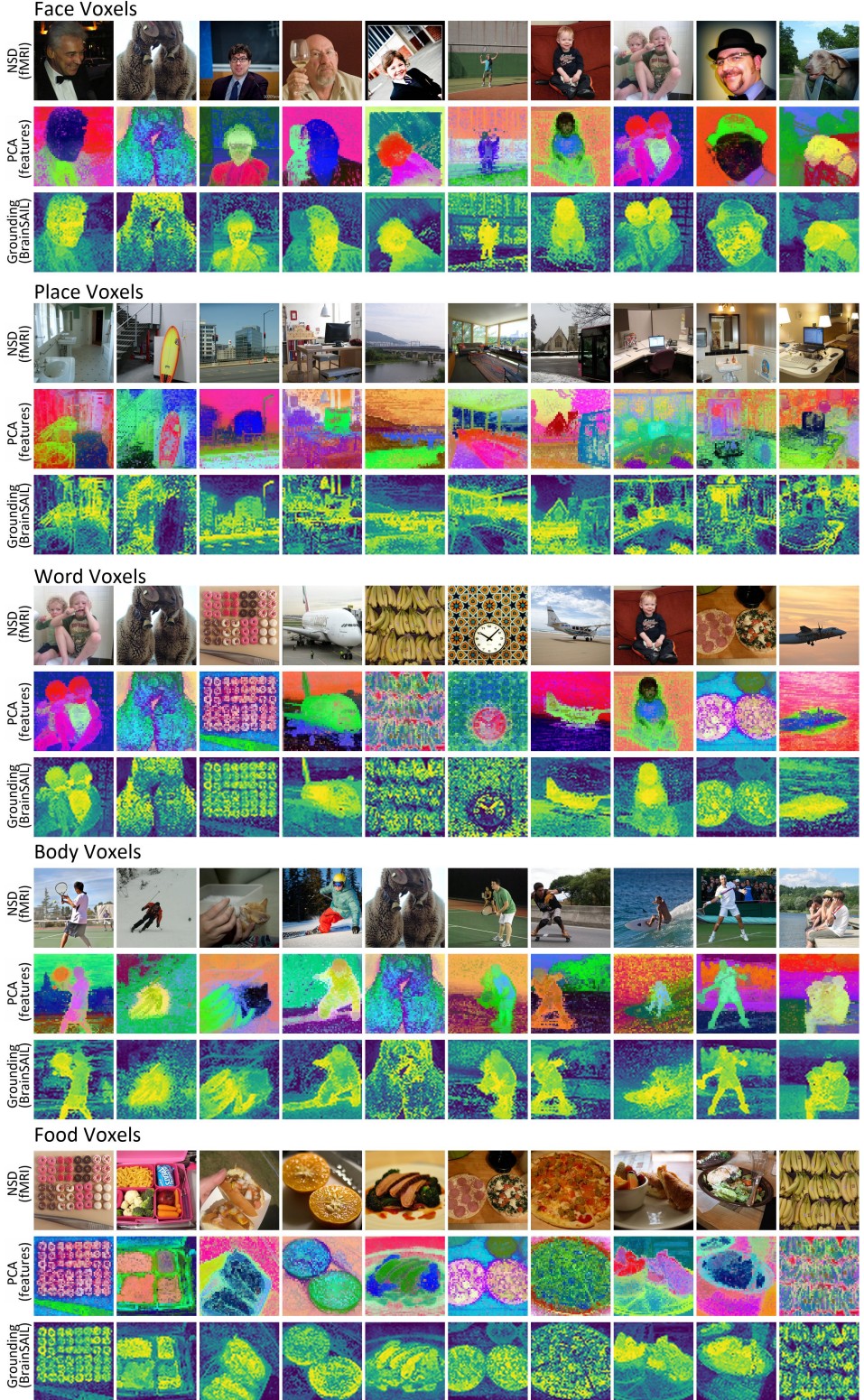

Figure S.18: **Grounding results using BrainSAIL for S2 CLIP backbone.**. We visualize the top test set images according to NSD fMRI for each category selective region. For each image, we also visualize the image-wise PCA for the distilled dense features. Note the PCA basis here is computed imagewise. For each image, we further visualize the feature relevancy map for the category selective voxels illustrating that this method extracts the semantically relevant regions in complex compositional images.

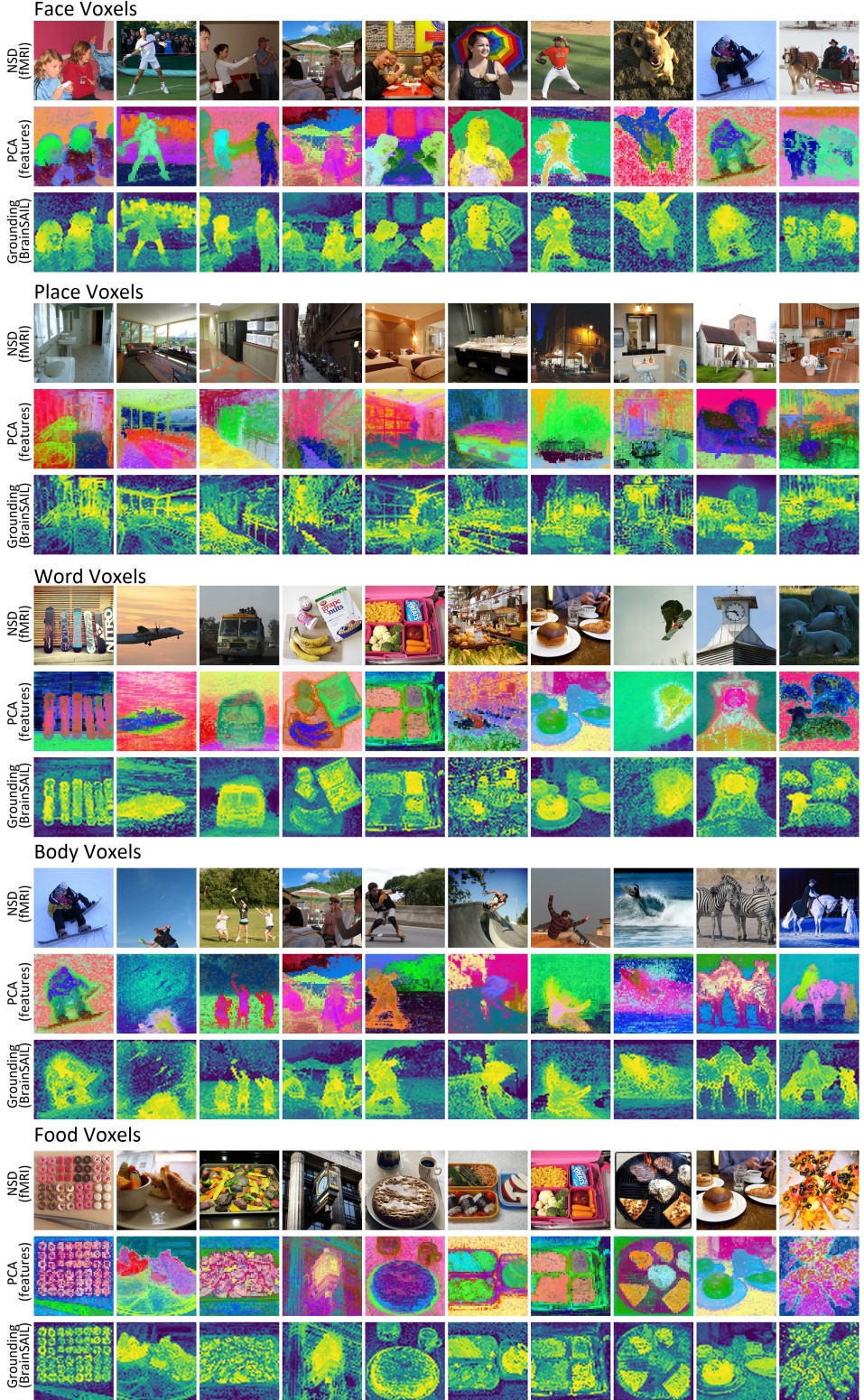

Figure S.19: **Grounding results using BrainSAIL for S5 CLIP backbone.**. We visualize the top test set images according to NSD fMRI for each category selective region. For each image, we also visualize the image-wise PCA for the distilled dense features. Note the PCA basis here is computed imagewise. For each image, we further visualize the feature relevancy map for the category selective voxels illustrating that this method extracts the semantically relevant regions in complex compositional images.

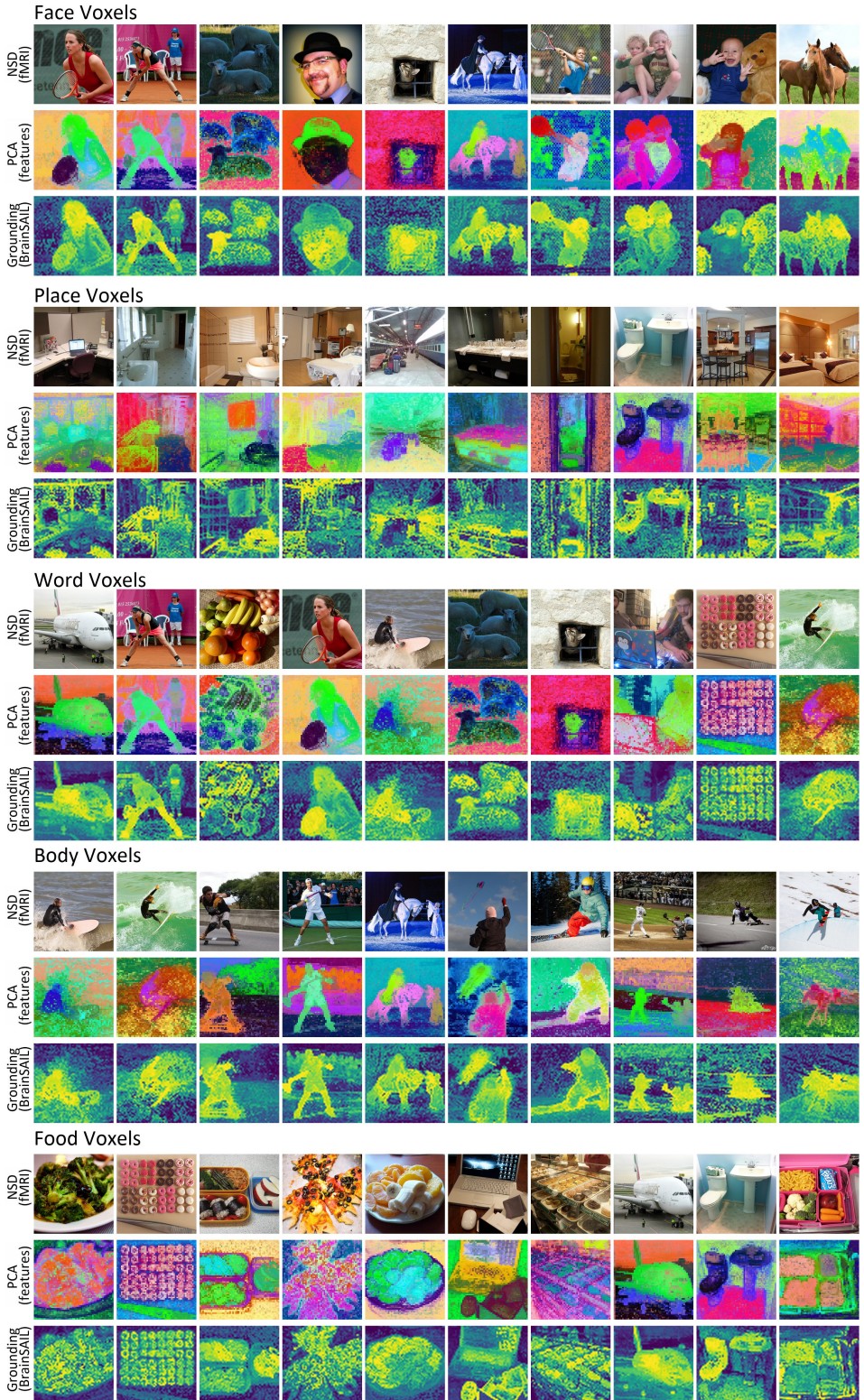

Figure S.20: **Grounding results using BrainSAIL for S7 CLIP backbone.**. We visualize the top test set images according to NSD fMRI for each category selective region. For each image, we also visualize the image-wise PCA for the distilled dense features. Note the PCA basis here is computed imagewise. For each image, we further visualize the feature relevancy map for the category selective voxels illustrating that this method extracts the semantically relevant regions in complex compositional images.

A.9   ADDITIONAL VISUALIZATION OF GROUNDING USING DINO BACKBONE

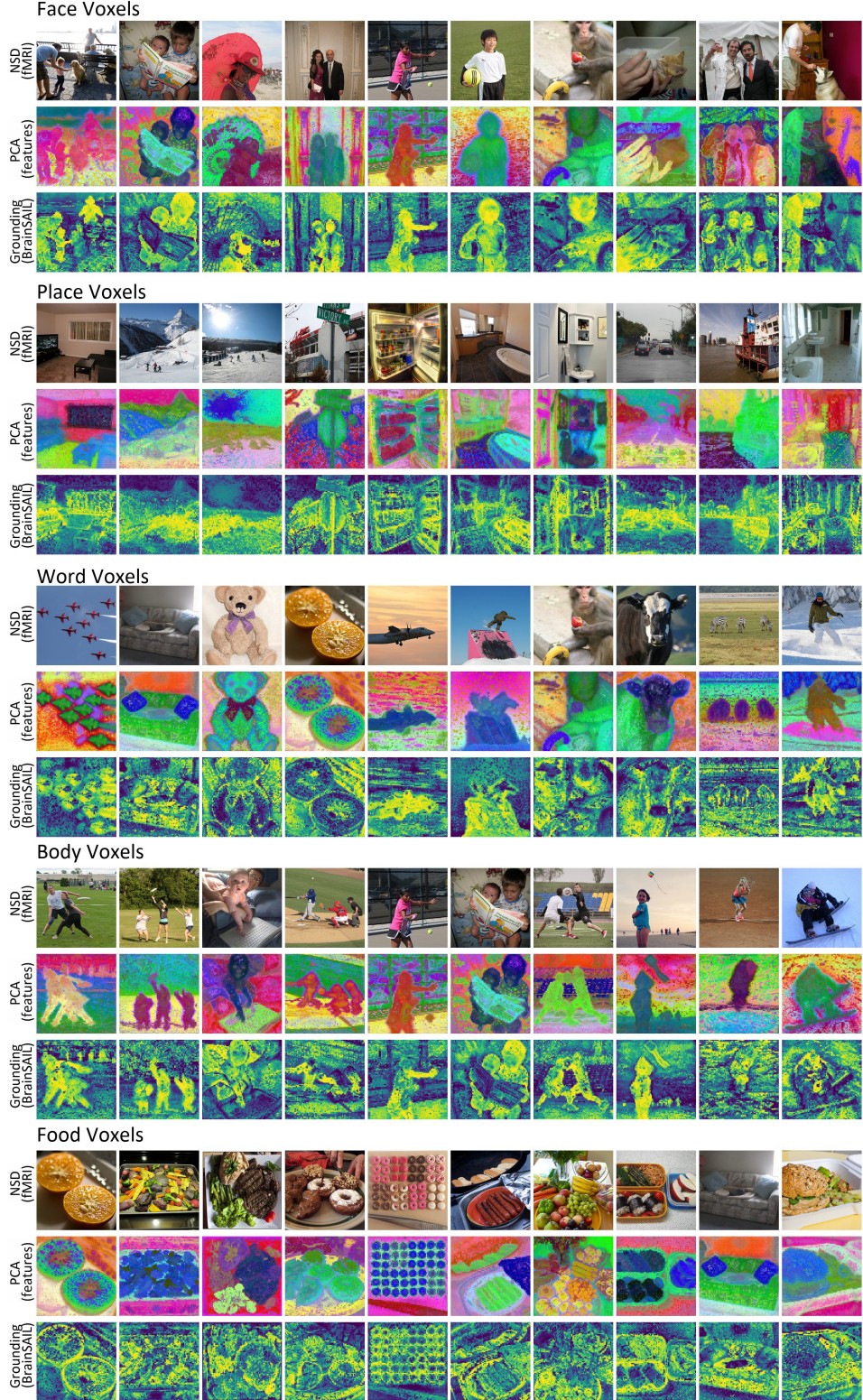

Figure S.21: **Grounding results using BrainSAIL for S1 DINO backbone.**. We visualize the top test set images according to NSD fMRI for each category selective region. For each image, we also visualize the image-wise PCA for the distilled dense features. Note the PCA basis here is computed imagewise. For each image, we further visualize the feature relevancy map for the category selective voxels illustrating that this method extracts the semantically relevant regions in complex compositional images.

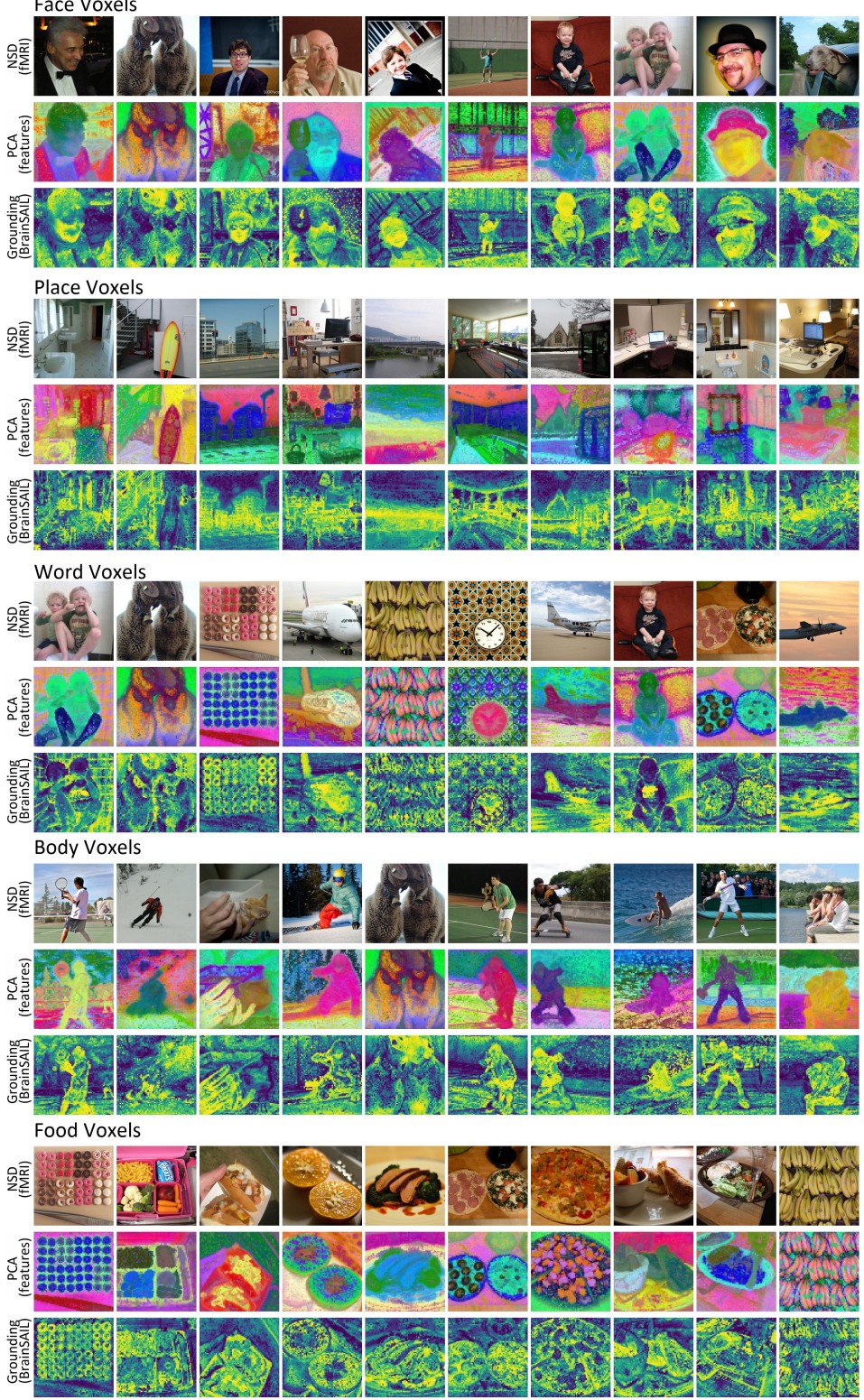

Figure S.22: **Grounding results using BrainSAIL for S2 DINO backbone.**. We visualize the top test set images according to NSD fMRI for each category selective region. For each image, we also visualize the image-wise PCA for the distilled dense features. Note the PCA basis here is computed imagewise. For each image, we further visualize the feature relevancy map for the category selective voxels illustrating that this method extracts the semantically relevant regions in complex compositional images.

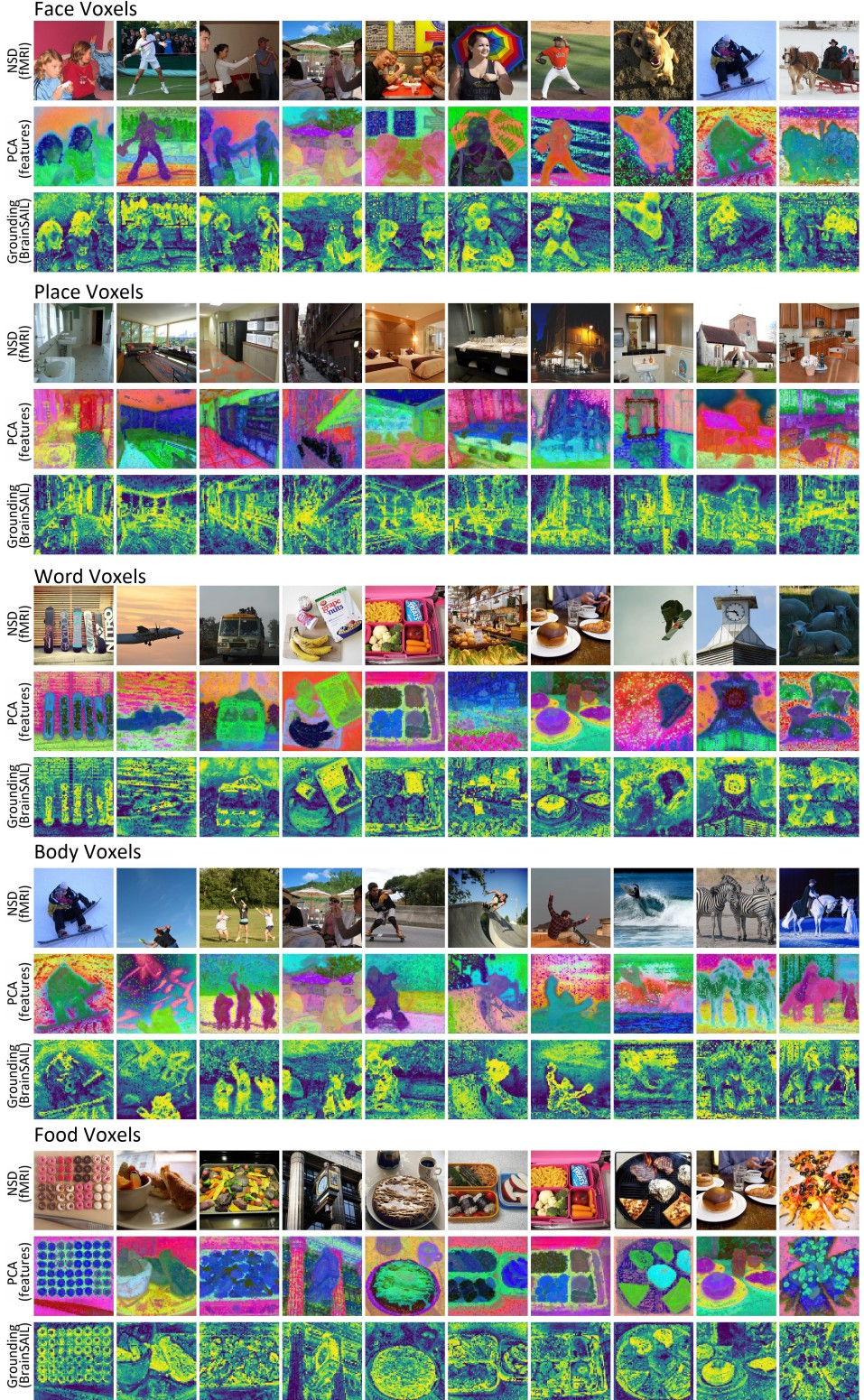

Figure S.23: **Grounding results using BrainSAIL for S5 DINO backbone.**. We visualize the top test set images according to NSD fMRI for each category selective region. For each image, we also visualize the image-wise PCA for the distilled dense features. Note the PCA basis here is computed imagewise. For each image, we further visualize the feature relevancy map for the category selective voxels illustrating that this method extracts the semantically relevant regions in complex compositional images.

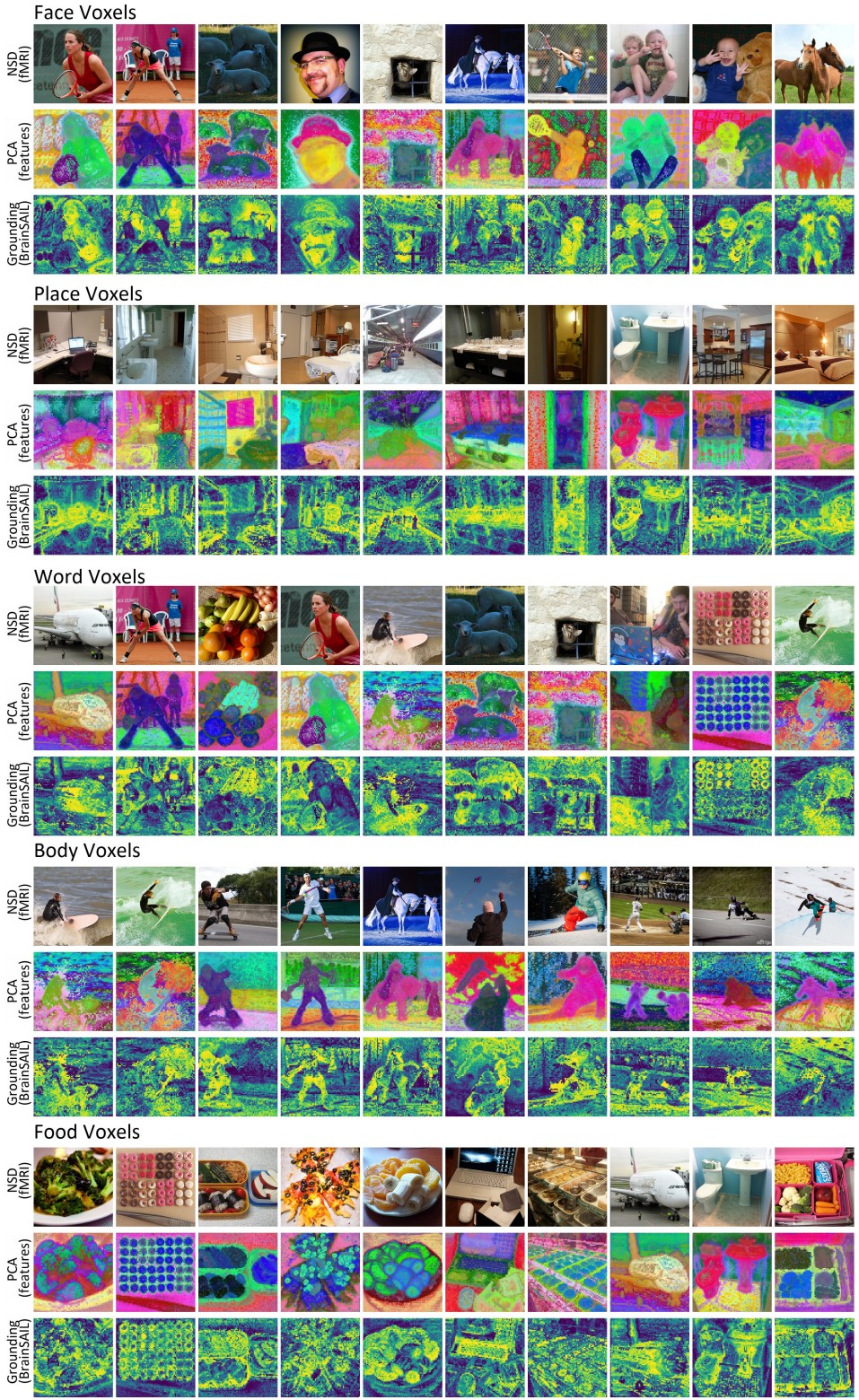

Figure S.24: **Grounding results using BrainSAIL for S7 DINO backbone.** We visualize the top test set images according to NSD fMRI for each category selective region. For each image, we also visualize the image-wise PCA for the distilled dense features. Note the PCA basis here is computed imagewise. For each image, we further visualize the feature relevancy map for the category selective voxels illustrating that this method extracts the semantically relevant regions in complex compositional images.

A.10 Additional Visualization of Grounding using SigLIP backbone

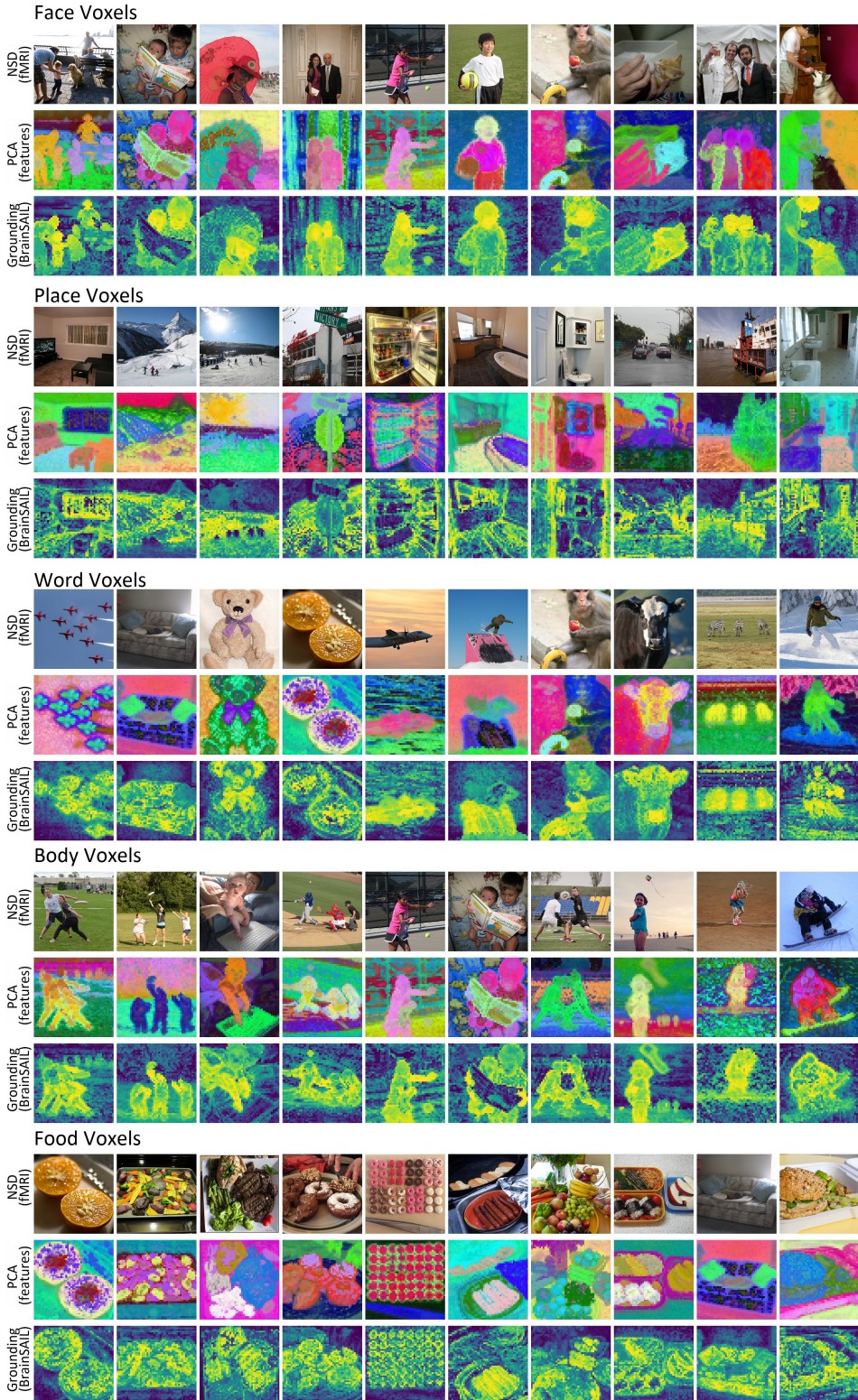

Figure S.25: **Grounding results using BrainSAIL for S1 SigLIP backbone.**. We visualize the top test set images according to NSD fMRI for each category selective region. For each image, we also visualize the image-wise PCA for the distilled dense features. Note the PCA basis here is computed imagewise. For each image, we further visualize the feature relevancy map for the category selective voxels illustrating that this method extracts the semantically relevant regions in complex compositional images.

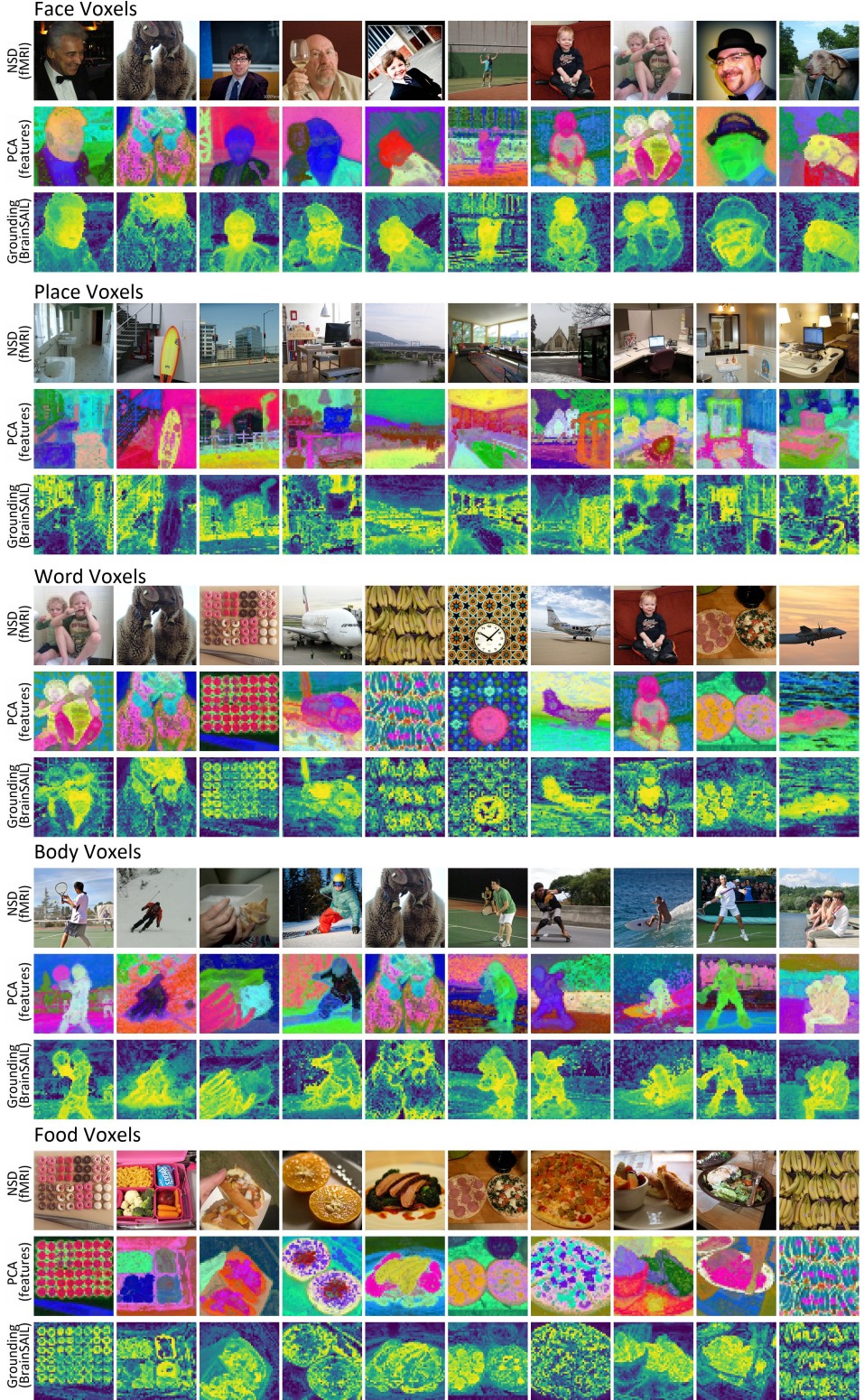

Figure S.26: **Grounding results using BrainSAIL for S2 SigLIP backbone.**. We visualize the top test set images according to NSD fMRI for each category selective region. For each image, we also visualize the image-wise PCA for the distilled dense features. Note the PCA basis here is computed imagewise. For each image, we further visualize the feature relevancy map for the category selective voxels illustrating that this method extracts the semantically relevant regions in complex compositional images.

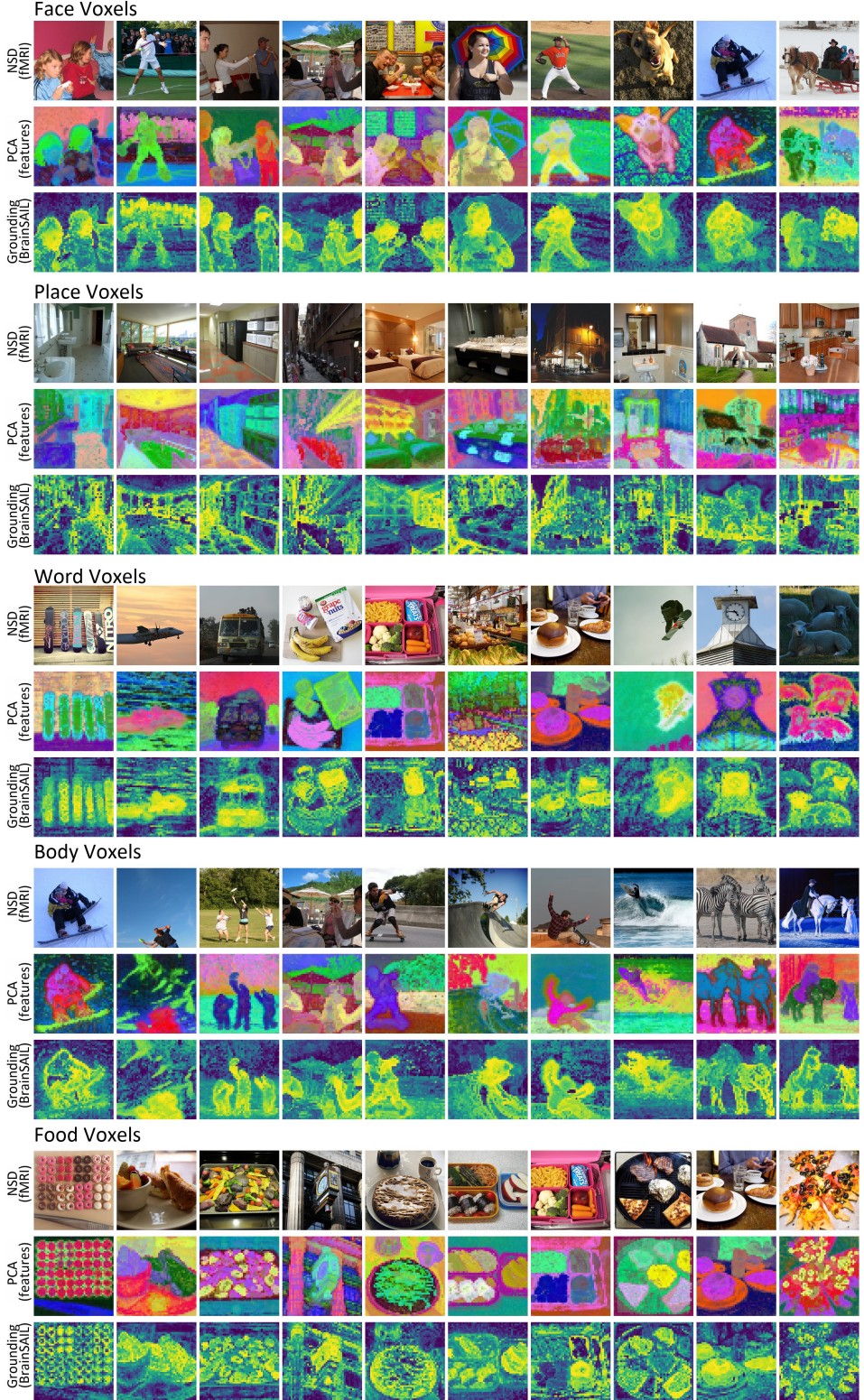

Figure S.27: **Grounding results using BrainSAIL for S5 SigLIP backbone.**. We visualize the top test set images according to NSD fMRI for each category selective region. For each image, we also visualize the image-wise PCA for the distilled dense features. Note the PCA basis here is computed imagewise. For each image, we further visualize the feature relevancy map for the category selective voxels illustrating that this method extracts the semantically relevant regions in complex compositional images.

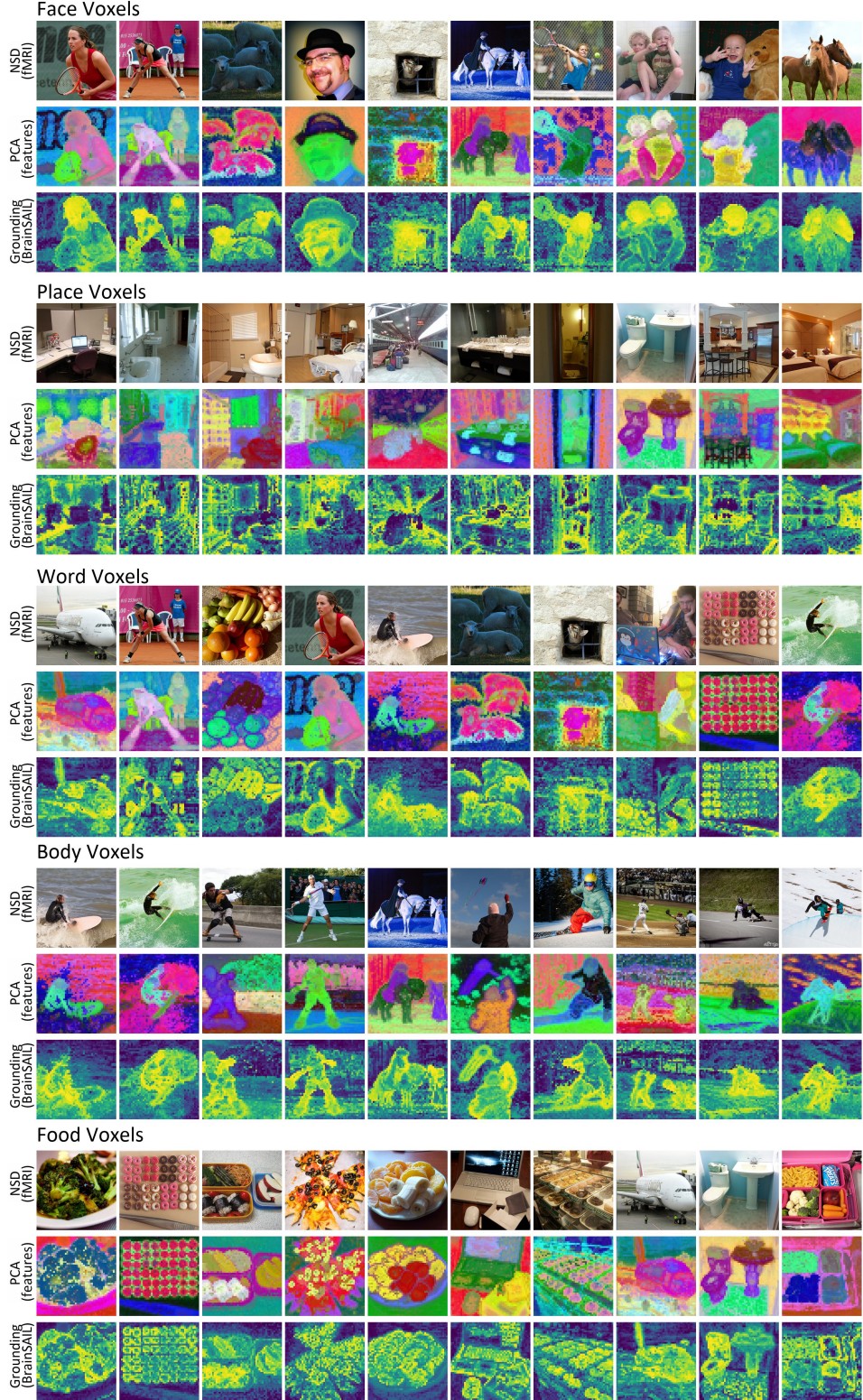

Figure S.28: **Grounding results using BrainSAIL for S7 SigLIP backbone.** We visualize the top test set images according to NSD fMRI for each category selective region. For each image, we also visualize the image-wise PCA for the distilled dense features. Note the PCA basis here is computed imagewise. For each image, we further visualize the feature relevancy map for the category selective voxels illustrating that this method extracts the semantically relevant regions in complex compositional images.

