# OpenReview forum: "Brain Mapping with Dense Features: Grounding Cortical Semantic Selectivity in Natural Images With Vision Transformers"
_ICLR.cc/2025/Conference — ICLR 2025 Poster_

### Official Review · Reviewer_PK4z · 2024-10-28

**Soundness:** 3
**Presentation:** 3
**Contribution:** 3
**Rating:** 6
**Confidence:** 2

**Summary:**

The paper introduces a method for mapping the brain activity obtained using fMRI to semantically meaningful features in image space using vision transformer backbones and a parameter-free distillation process for denoising.

**Strengths:**

- The method was explained clearly
- Literature review seems exhaustive
- Claims SOTA (although I couldn't verify this)

**Weaknesses:**

-Abstract strats with features of BrainSAIL (first few sentences) before getting into WHAT BrainSAIL is.

-It is not clear to me after training the backbone and distillation, how much of the output result is actually driven by brain data or completely rely on features of the image learned by the frozen foundation model.

-SOTA was claimed but I wasn't able to find any comparison to other relevant methods in the paper.

**Questions:**

How is it possible to verify that the method is not dominated by the foundation model representation of image rather than actually representing brain semantic representations?

---

> ### Author Response · Authors · 2024-11-21
> **Response to Reviewer PK4z**
>
> We thank **Reviewer PK4z** for the concrete and helpful suggestions! We are strongly encouraged by your evaluation that our method is clearly explained. We appreciate your advice on clarification of the model versus brain contributions.
>
> We address specific comments below and update the revision with additional results.
>
> Per your suggestion, we have also modified the first few lines of the abstract to specifically start with an explanation of what BrainSAIL is.
>
> >**Q1) SOTA results for open-vocabulary semantic segmentation**
>
> We consulted with both the authors of SCLIP (ECCV 2024) and the authors of NACLIP (WACV 2024) while running our evaluation. NACLIP is itself an improvement of SCLIP and claimed SOTA open-vocabulary segmentation results on a majority of datasets earlier this year. Our evaluation results are shown in **Table 1** of the main text.
>
> We followed their papers and perform quantitative evaluation on four of most widely used segmentation datasets -- `PASCAL VOC 2012`, `ADE20k`, `COCO-stuff`, and `COCO-Object` in ***Table 1*** of the main paper. All three methods (SCLIP, NACLIP, and our method which leverages distillation) are evaluated without PAMR refinement according to the suggestion in the SCLIP paper. No PAMR is more realistic as it does not require discrete labels, and also does not assume a spatial prior over labels. We independently ran the code and used the same evaluation harness for all three models.
>
> We were able to confirm that NACLIP is indeed an improvement over SCLIP in our own experiments.
>
> We evaluate two metrics:
> 1. The first -- mean Intersection over Union (mIoU) metric is commonly used in segmentation, but requires knowledge of the ground truth classes during test time to compute an argmax.
>
> 2. The second metric is the Pearson correlation between CLIP-based saliency maps (computed using the CLIP-text embedding cosine similarity with dense image embeddings) and ground-truth labels. This avoids assumptions about specific categories, and is a more realistic reflection of our use case.
>
> We consistently improve over NACLIP in all four datasets and both mIoU and correlation metrics.
>
>
> >**Q2) Contribution of the backbone and the brain**
>
> We would like to clarify that our work explicitly uses the representations of the backbone (CLIP/DINO/SigLIP or any other vision transformer foundation model) to identify the regions that activate a given brain region.
>
> We agree that it would be helpful to see how representations differ between vision foundation models (for example what can CLIP identify in an image) and what the brain selects as relevant for a given brain region (for example face or body region in the brain).
>
> * We observe that CLIP dense features can have very fine-grained semantic attributions when probed with the CLIP text encoder, and is able to localize arms/legs/hair/eyes/ears/nose/mouth and other body parts for humans & animals.
>
> * We do not observe consistent separation between face and body voxels when probed using either fMRI voxel-wise or region-of-interest weights. This is likely due to the nature of the fMRI stimulus set itself. Specifically, examination of the NSD images indicates a significant co-occurrence of faces and bodies within the stimuli. Our strategy of freezing the fMRI encoder backbone mitigates this co-occurrence issue, compared to training the entire network as in brain dissection **[1]**.
>
> We have added additional results on Page 20 of the revision (section A.2 of the supplemental, highlighted in blue).
>
> >**Conclusion**
>
> In response to your suggestions, we have added additional experiments to the revision clarifying the difference in representation granularity between the vision model and the brain and revised the abstract. We hope our response has been helpful, and you could consider a more positive evaluation of our work!
>
> **References**
>
> **[1]** Brain dissection: fMRI-trained networks reveal spatial selectivity in the processing of natural images (*NeurIPS* 2023)

---

> > ### Author Response · Authors · 2024-11-26
> >
> > Hello Reviewer PK4z,
> >
> > Do you have any remaining questions or suggestions after reviewing our response? Please feel free to share them with us!
> >
> > We are happy to answer any questions and help in any way we can in the time remaining.

---

> > ### Comment · Reviewer_PK4z · 2024-11-26
> >
> > Thanks for clarification and edits to the paper. I raised the score.

---

### Official Review · Reviewer_dUkJ · 2024-11-02

**Soundness:** 3
**Presentation:** 3
**Contribution:** 3
**Rating:** 6
**Confidence:** 4

**Summary:**

The authors proposed a method called BrainSAIL to deal with the problem of multiple categories co-existing in the natural images, combining the image embeddings and dense visual features and further identifying specific sub-regions of each image, which reflects the selectivity patterns in different areas of the higher visual cortex. BrainSAIL can realize the semantic attributes and positioning of related objects in complex images, which helps decompose the high-level visual representation of the human brain.

**Strengths:**

1.	BrainSAIL can obtain dense embeddings without artifacts and additional training through an efficient learning distillation module for high-throughput presentation of the visual cortex on large datasets.
2.	Compared to other studies that used simplified images, BrainSAIL can isolate specific image regions that activate different cortical areas when viewing natural scenes, providing a more complete description of how real-world visual stimuli are represented and processed.
3.	The experiments are very sufficient, especially in the appendix.
4.	This work demonstrated that an artifact-free dense feature map can be derived to explore high-throughput selectivity in the visual cortex.

**Weaknesses:**

1.	This method has a high model dependency (e.g., CLIP, DINO, SigLIP). While this improves efficiency, it limits flexibility to adapt feature extraction to the specific needs of neural regions, potentially constraining the granularity of semantic selectivity.
2.	The method's fMRI training uses specific datasets (e.g., NSD), which may introduce bias, limiting generalizability across different populations or visual tasks—especially when the dataset’s images or semantic information don't cover the full range of possible visual stimuli.

**Questions:**

DINO trained without language guidance shows high sensitivity to low-level visual features, does this hurt its performance on high-level semantic selection?

---

> ### Author Response · Authors · 2024-11-21
> **Response to Reviewer dUkJ**
>
> We thank **Reviewer dUkJ** for the positive evaluation of our work and the concrete suggestions! We will address your suggestions below and include an additional discussion in the revision regarding the limits imposed by NSD.
>
> > **Q1) Flexibility of backbone choice**
>
> We would like to clarify that our method could be used with any vision transformer based model. As far as we are aware, most vision foundation models utilize a vision transformer architecture (including OpenAI's CLIP, Google/Nvidia's SigLIP/RADIO implementation, and Meta/Facebook's DINO). Recent research by Wang et al. **[1]** and Colins et al. **[2]** have also identified CLIP ViT as the highest performing predictor of higher visual cortex activations using fMRI.
>
> Due to the flexibility of our method, our method is able to work with the vast majority of state-of-the-art vision backbones proposed recently.
>
> We agree dynamically picking the best backbone according to the voxel is an interesting research direction. As shown by **[1]**, different models may be optimal for different brain regions. We believe this is an interesting future research direction.
>
> > **Q2) Diversity of NSD**
>
> We completely agree that our results depends on the stimulus set diversity. We utilized NSD as it is currently the highest-quality (7T, three repeats per image) and largest (around 10,000 images per subject) fMRI dataset.
>
> As mentioned by **Reviewer PeSm**, the diversity of NSD images has been raised as a potential issue by Shirakawa et al. (Kamitani senior author) **[3]**.
>
> We do want to point out that our method is an improvement over Brain Dissection (NeurIPS 2023). Their method trained a response optimized convolutional network over ~10,000 images. This means that the fMRI encoder network would be strongly constrained by the semantic/visual biases present in the images, and also inherit the local feature bias present in ConvNets.
>
> In contrast, our work is able to leverage the representations learned by these vision foundation models over hundreds of millions of images, which mitigates the bias caused by the fMRI dataset.
>
> Following yours and **PeSm**'s suggestion, we have added additional discussion in the revision discussing the limitation.
>
> >**Conclusion**
>
> We agree with you regarding the issues posed by the limited diversity of the NSD stimulus set. We use this dataset as it is currently the largest and highest quality fMRI image viewing dataset. Our method partially alleviates the problems posed by the limited dataset diversity compared to Brain Dissection (NeurIPS 2023).
>
> We have updated our revision to include additional discussion regarding dataset diversity. We hope our response has been helpful, and you could consider a more positive evaluation of our work.
>
> **References**
>
> **[1]** Better models of human high-level visual cortex emerge from natural language supervision with a large and diverse dataset (*Nature Machine Intelligence* 2023)
>
> **[2]** A large-scale examination of inductive biases shaping high-level visual representation in brains and machines (*Nature Communications* 2024)
>
> **[3]** Spurious reconstruction from brain activity (*arXiv* 2024)

---

> > ### Author Response · Authors · 2024-11-26
> >
> > Hello Reviewer dUkJ,
> >
> > Do you have any remaining questions or suggestions? We would appreciate your input!
> >
> > We would be happy to clarify and help out in any way in the remaining time.

---

> > > ### Comment · Reviewer_dUkJ · 2024-11-26
> > > **To author**
> > >
> > > Thanks for your repsones. This rebutall addresses all of my concerns. I will keep my score.

---

### Official Review · Reviewer_PeSm · 2024-11-03

**Soundness:** 4
**Presentation:** 4
**Contribution:** 4
**Rating:** 8
**Confidence:** 4

**Summary:**

This paper aims to explore the semantic topography in human visual cortex using fMRI. The main goal is to disambiguate the typically confounded nature of multiple categories which is a feature/issue in naturalistic images. The trend towards naturalistic datasets in recent years necessitates more advanced methods to be able to deal with such confounds and this work takes a good step forward in the right direction. This paper then goes on to look at what regions of human visual cortex are driven by low-level image features, providing a complementary view into how the authors’ method can be used beyond studying categorical selectivity. This method provides a good solution to explore such potential confounding effects in determining what is driven by high or low level visual information.

It was a very nice paper to read and shows the authors' clear expertise and attention to detail in terms of the questions we should be asking in the community.

**Strengths:**

- The authors show alternatives to high-frequency artefact removal that are more computationally efficient than other solutions out there, namely by recognising the existing common method of adding register tokens in ViTs can be replaced with their proposed learning-free distillation module, which avoids the need for (computationally demanding) additional training.
- The authors show that this method is a good way to target high confounds in naturalistic images by studying things like colour confounding with food images, luminance modulations in inside/outside scenes, depth information in varying images containing varying “reach spaces” etc.
- The paper highlights potential equivalence between representations derived under varying training algorithms and questions what is being learned and the generalisation of features. I found this to be timely as recent discussions with LLMs and the utility of methods like BrainScore have been brought into question. I think we need to be asking these questions across the board and I appreciate this final section of the paper that raises this issue.
- Visualisations are very nice and clear, making some more computationally dense text a bit more clear with reference to specific examples that underlie various findings (e.g. the effects of no language supervision in DINO and the downstream affects on the derived image maps)

**Weaknesses:**

- PDF is nearly 50 pages long. While ICLR allows for 10 main pages and unlimited supplementary information, the density of what is provided is a bit on the extreme side. In light of this, I still found myself searching for implementational information that I felt was missing and could be added in a revised version (see *Questions*)
- I reformulated many of the comments I first wrote as weaknesses into more directed questions in the section below, hoping that these points might be more easily addressable and will hopefully make it into the camera-ready version such that readers have a better experience of the paper by benefiting from some of the points of clarity that I ran into while reading.

**Questions:**

- “We validate this dense feature mapping method on a large-scale fMRI dataset consisting of human participants viewing many thousands of diverse natural images that span a wide range of semantic categories and visual statistics (Allen et al.,2022).”
    - The work by Kamitani recently showed that UMAP gave something like 40 clusters of semantic diversity at a broad level, forcing us to re-evaluate our assumed understanding of just how semantically varied NSD actually is. Do you believe the findings in the "spurious reconstructions" paper have an implication on the notion of semantic diversity as described in your work?

- The introduction has repeated references to BrainSAIL’s dense embedding framework but I’m not sure why this point is continually brought up as the vision model embeddings on the natural image are dense to begin with and I can’t see that the implementation of a step to make embeddings more dense is applied anywhere, so I was a bit lost on this point. You have the adapters and a linear probe and everything always seems to be in a dense (not non-sparse) setting. Some clarification on this point would be welcome in the paper as you seem to mean something different to how I understand the same phrasing.

- Was there supposed to be a section explaining the data choices for the brain data (surface vs voxels, subject-space vs template-space) etc.? This is kind of left to be assumed from other sections of the data but I think a small section in the supplemental outlining the data usage choices more specifically would render the procedure more complete with regard to necessary experimental details. For example, the fact you’re using the masks in Jain et al. (2023) implies working in standard (normalised) template space, but I sort of expected being able to verify this explicitly via some technical description somewhere in the supplemental materials.

- You described how you selected varying functional regions from the dataset (food masks in Jain et al. (2023) and t > 2 in the Stigliani fLoc dataset) but the selected voxels being plotted, the selection of them, is not described and also a piece of missing information. Was this taken by looking at noise ceiling values calculated in NSD and if so, what value was used as the threshold?
- Lines 427-429: you identify the areas surrounding the FFA to be selective for colour, and point out this corresponds to the food area previously identified by others. Can you elaborate on the apparent contradiction between your assumption that this is actually a food area (identified earlier in the paper) or is it one driven by high colour-sensitivity and therefore might be confounded with highly colourful food images?
- Figure 7: I don’t understand how these values are calculated and the y-axis is not labelled. I wanted to ask you to be more precise on what “spatial similarity” means in terms of the method use to derive the results.
- Not quite sure the logic for why the procedure described with the learning-free distillation module results in denoised features. Why does applying a transform, running an image through an encoder, then projecting the results back again, result in denoised features? It seems to work but I just felt like the method was explained in the "how" and not "why" it works. The paper could be made stronger by adding in some intuition in the relevant section.

**Details Of Ethics Concerns:**

N/A. Public Dataset used.

---

> ### Author Response · Authors · 2024-11-21
> **Response to Reviewer PeSm [Part 1 of 3]**
>
> We are grateful for your suggestions and detailed review! We will address specific questions below, and will incorporate all of your feedback into the revision.
>
> > **Q1) Structure of the paper supplemental**
>
> We have reordered the supplemental in the revision to bring forward the technical details according to your suggestion. We have also included additional implementational specifics to clarify our results and methods (blue text).
>
>
> > **Q2) Limitations of NSD dataset diversity**
>
> We agree with the overall conclusions of "Spurious reconstruction from brain activity" **[1]** by Shirakawa et al. (Kamitani senior author). We have now cited it in the revision. We used NSD as it is currently the largest and highest quality static image viewing fMRI dataset.
>
> We believe that a semantically & visually diverse fMRI image stimulus set would lead to more robust conclusions for both decoding models and selectivity analysis. A motivation of our BrainSAIL work was to address the limitations present in "Brain Dissection" (NeurIPS 2023), which constructed a task-optimized fMRI encoder by training a ConvNet only on the ~10,000 subject-wise stimulus images. By constraining the training set to just the NSD stimulus set and using a CNN, the network would be strongly affected by the semantic biases and diversity limitations of the stimulus set and the local inductive biases imposed by CNNs **[2]**.
>
> We instead take frozen vision transformer backbones (CLIP/DINOv2/SigLIP), and leverage their training data diversity (hundreds of millions to billions of images each) to partially mitigate the NSD biases and CNN biases.
>
> Regarding the specific Fig 4 in Kamitani, we want to note their figure was based on `CLIP text embeddings` of COCO captions of NSD images. We did not see such a strong discrete clustering effect on `CLIP image embeddings` when we tried a UMAP plot. This suggests that text may have a discrete bottleneck-like effect, and omit some of the semantic variations present in the full image. But we do agree that increased diversity would be helpful.
>
>
> > **Q3) Dense embedding framework**
>
> Our framing was more in terms of how people currently train and use vision foundation models in the context of ML & fMRI encoders.
>
> Contrastively trained vision transformers (CLIP/DINO) or contrastive-esque models (SigLIP), have dense features that do not natively lie in the same semantic space as the `[CLS]` token. This phenomenon was first studied by the "Extracting Free Dense Labels from CLIP" paper.
>
> This issue can be mitigated when vision transformers are used in vision-language models. In this use case, a translation networks is typically trained between the vision & language model, alongside training of the language model itself. Because full training is possible, it does not matter that the dense features are not semantically aligned with the`[CLS]` token.
>
> Prior work (feature field distillation -- NeurIPS 2022) address the problem of dense/sparse misalignment by using LSeg (ICLR 2022), and obtained clean semantic embeddings via neural field distillation via multiview consistency. This approach has three problems:
>
> **(1)** As has been noted by LERF (ICCV 2023) and our own observation, LSeg has very poor characterization of objects not in the training set. We do not wish to restrict our investigation of fMRI selectivity only to objects in existing segmentation datasets.
>
> **(2)** Unlike in 3D, we do not inherently have multiview data to denoise semantic embeddings.
>
> **(3)** Neural field learning and inference are both prohibitively slow (hours per scene for learning), which renders investigation of tens of thousands of voxels in higher visual cortex over thousands of NSD stimulus images impossible.
>
> To address these issues we propose the following:
>
> **(1)** Use of recent proposed self-attention modifications (MaskCLIP or NACLIP) to align the dense`[CLS]` embedding spaces. These modifications preserve the semantic representation of vision models trained on hundreds of millions of images. We no longer have to rely on LSeg which finetunes on very small supervised segmentation datasets.
>
> **(2)** We generate spatial-offsets & flips of existing images to generate "multiple views". We show that this yields an effective denoising of the dense embeddings.
>
> **(3)** By leveraging an "inverse transform" which operates by un-shifting and un-flipping the dense embeddings, we show that mean aggregation yields the same mathematical solution as neural fields learned with MSE and gradients.
>
> We evaluate this in Table 1, and show that our dense embeddings achieve state-of-the-art segmentation results on the major segmentation datasets. In response to your suggestion, we have added a discussion on **Page 19 (section A.1)** of the revision discussing why our method works.
>
> **Response continued in next section**

---

> ### Author Response · Authors · 2024-11-21
> **Response to Reviewer PeSm [Part 2 of 3]**
>
> > **Q4) Selection of voxels to model**
>
> We would like to emphasize that the dense features are computed in a way that they can be re-used to model any voxel.
>
> The voxel selection only plays a role in the encoder training (linear weights on a frozen backbone), which can be done relatively quickly if a new set of voxel masks were chosen.
>
> We select voxels which could be broadly defined as "higher visual cortex". This selection is based on the average noise ceiling within an HCP parcellation. We compute the average noise ceiling for each HCP region in each subject, and compute a ranking of the regions within each subject. We average the ranks across the subjects, and select the top 25% regions (45 out of 180 HCP). We then exclude voxels which are listed as "early visual" (V1~V4) according to the NSD pRF experiment or under the HCP definitions. This is a relatively generous thresholding.
>
> We find this procedure reliably picks out visual cortex voxels without explicit constraints.
>
> For visualization purpose in Figure 3, we do not explicitly select voxels based on functional localizer experiments, and the UMAP is not done by conditioning the procedure with category data.
>
> We have further clarified this in page 19 (**section A.1**) of the supplemental (blue text).
>
> > **Q5) Feature correlates**
>
> We thank the reviewer for highlighting the complex issues surrounding the representation of both color and food in human visual cortex (the same question applies to other domains, such as curvature and faces, or texture and tools). This is an issue we have considered extensively.
>
> At the present time, there is no clear answer to the relationship between food-driven neural responses in this region and color-driven responses in roughly this same region. What we can say with some confidence is that, as shown in Figure 5, the maps derived from our spatial attribution framework show a correlation between the region surrounding the FFA and high color saturation areas in the images. And to be clear, our current findings cannot disambiguate the contributions of color and food selectivity in these responses.
>
> We believe that labeling a brain area the "X area" reifies the observation that it responds more strongly to images of X than other categories. With respect to the specific question about the relationship between the "food area" and color, we believe this is an open question. There are several possible interpretations of the overlap between food-driven and color-driven neural responses. One possibility is that this overlap reflects an association between warm, saturated hues and the presence of food. However, a recent analysis revealed that color saturation and warmth are encoded independently of food **[3]**, consistent with color metrics only accounting for a portion (sic) of the response variance in food-selective visual areas **[4]**. Further demonstrating independence between food and color, an experiment comparing the visual responses of food and non-food images found that food-selective regions are also identifiable using decontextualized, grayscale images **[5]**. We hypothesize that although color is not a prerequisite for food-selective visual responses, color may contribute to inferences about substance or material and is, therefore, co-located with local representations of food. Consistent with this point, the left medial fusiform gyrus (collateral sulcus), which is roughly the same region of visual cortex that has been identified as sensitive to both color and food in our present paper and in **[3] [4] [5]**, also encodes features related to surface texture and material, both thought to be important in grasping and manipulating objects **[6]** **[7]**.
>
> We acknowledge using the shorthand "food area" convention which can be misleading because it is often difficult to surface all of these issues. We have clarified in a revised version of our paper by using the term "food responsive".
>
> **Response continued in next section**

---

> ### Author Response · Authors · 2024-11-21
> **Response to Reviewer PeSm [Part 3 of 3]**
>
> > **Q6) Axis label**
>
> This was an oversight on our part. Figure 7 is measuring the spearman correlation of the saliency maps between two methods for the top-100 ground truth most activating images from the fMRI test set for a given region. A higher value indicates two models have attribution maps that are more similar. 1 indicates perfect correlation, 0 indicates no correlation, and -1 indicates negative correlation.
>
> We have updated this figure in the revision according to your suggestions.
>
> > **Q7) How does denoising work?**
>
> We were motivated by "Feature Field Distillation" (NeurIPS 2022) which proposed to solve a similar problem in 3D. We discuss some background in our response in **Q3**.
>
> To further expand upon this, multiview consistency training via neural fields has been shown to effectively remove measurement noise in 3D via neural radiance fields or feature fields.
>
> In 2D, we do not have multiview images, so we offset and flip an image to simulate multiple views. The augmentations that we choose do not modify the semantics of an image -- a red bird is still a red bird even if flipped left/right or viewed slightly offset. We have observed that the artifacts are not fixed in space relative to scene content. So by offsetting and flipping an image, we can effectively change these artifacts into measurement noise.
>
> The inverse transform is implemented via a spatial location lookup to identify the original location of a patch before offset/flipping.
>
> We have added a discussion on **Page 19 (section A.1)** of the revision discussing our motivations for constructing the denoising method this way.
>
>
> > **Conclusion**
>
> We thank **Reviewer PeSm** for the helpful suggestions. We hope that our responses have been helpful. We have updated our paper by reorganizing the supplemental, including additional implementation details, and providing additional motivation and intuition.
>
> **References**
>
> **[1]** Spurious reconstruction from brain activity (*arXiv* 2024)
>
> **[2]** Do Vision Transformers See Like Convolutional Neural Networks? (NeurIPS 2021)
>
> **[3]** Color-biased regions in the ventral visual pathway are food selective (*Current Biology* 2023)
>
> **[4]** A highly selective response to food in human visual cortex revealed by hypothesis-free voxel decomposition (*Current Biology* 2022)
>
> **[5]** Selectivity for food in human ventral visual cortex (*Nature Communications Biology* 2023)
>
> **[6]** Domain-Specific Diaschisis: Lesions to Parietal Action Areas Modulate Neural Responses to Tools in the Ventral Stream
> (*Cerebral Cortex* 2019)
>
> **[7]** Domain-specific connectivity drives the organization of object knowledge in the brain (*Handbook of Clinical Neurology* 2022)

---

> > ### Author Response · Authors · 2024-11-26
> >
> > Hello Reviewer PeSM,
> >
> > Do you have any remaining questions or suggestions in light of our response? We welcome your input!
> >
> > We hope to clarify any points you wish to discuss during the remaining time.

---

> > > ### Comment · Reviewer_PeSm · 2024-11-26
> > > **Nice response**
> > >
> > > I thank the authors for the time they put into these responses. They brought up many further issues that were not clear to me during the reading of the paper, issues which explain choices made in the paper. These further show the care and attention put into this work. I maintain my current rating (which is a clear recommendation of acceptance).

---

### Official Review · Reviewer_yyM7 · 2024-11-04

**Soundness:** 3
**Presentation:** 3
**Contribution:** 3
**Rating:** 6
**Confidence:** 3

**Summary:**

Methods:
- Embeddings from deep neural networks are used to predict fMRI recorded brain-responses.
- Features are extracted from the model and input to a denoising process. The smoothed features are then input to the fMRI encoder (I think) to create spatial attribution maps.
- UMAP is applied to the encoder parameters, which uncovers previously reported category selective regions for faces, places, words, food, and bodies.
- The spatial attribution maps are correlated with pixel depth, color saturation, and color luminance to create voxel-wise correlation values for these image properties. Place areas were found to be more correlated with depth, food areas with color, and OPA with color/luminance.

**Strengths:**

The paper is well written and clear. The main contribution is the use of model features to create voxel and image-wise spatial contribution maps. This is quite useful as an interpretation technique. The use of pixel-wise metrics like depth, saturation, and luminance are a powerful extension of the method.

**Weaknesses:**

All of the investigated category-selective areas (food, places, words, faces, bodies) are previously reported. Would be interesting so see this tested on less common or more fine-grained categories.

I found the explanation of the methods is a bit confusing (see questions). Could use clearer high-level descriptions before diving into the finer details.

I think there should be a visual comparison of attribution maps for different voxel categories with the same images. I was able to compare a few in figure 3 since there are some repeated images in words/food and face/body voxels. If the method is working then the word voxels should highlight words more than voxels for other categories. I would expect to see the face voxels highlighting faces a lot more than body parts, and the opposite for body voxels. However in the picture of the baby shared between the body/face voxels I don't see much of a difference in their attribution maps.

**Questions:**

1. My understanding is that the voxel-wise adapters predict a scalar brain response from an M-dimensional output of the backbone model. How is the voxel-wise adapter then applied to the dense features? I am thinking the dense features have a shape (H, W, M), and the adapter transforms the last dimension to get a scalar (H, W) activation map (figure 1.b). I found this a bit unclear in the paper.

2. The learning free distillation is applying random spatial transformations to the input image, applies the inverse transformation to the features, and then averages them all together. Is my understanding correct?

3. I do not understand what equations 3-5 are about.

4. Figure 4 states that the UMAP basis is computed image-wise. I found it unclear how this was done, and what is the significance of an image-wise UMAP?

---

> ### Author Response · Authors · 2024-11-21
> **Response to Reviewer yyM7**
>
> We thank Reviewer yyM7 for the helpful review and positive evaluation of our work! We address your suggestions below, and update the revision to provide further clarification.
>
> > **Q1) Categories not included in functional localizers**
>
> We would like to clarify that voxels for **Figure 3** were based on a broad definition of "higher visual cortex", and were not selected based on functional localizer information.
>
> For **Figure 4** -- due to our use of the NSD dataset, we only have access to functional localizer masks for faces/places/bodies/words, and access to subject-space food masks from **[1]**.
>
> We did perform a feature correlate search on a voxel-wise basis using depth/color saturation/color luminance in **Figure 5**, and find that we were able to indeed identify scene selection regions (RSC/OPA/PPA), food selective regions adjacent to FFA, and indoor/outdoor selective sub-regions in OPA without region-wise assumptions.
>
> > **Q2) attribution maps for different voxel categories with the same images**
>
> Regarding the baby picture shared between body/face voxels. We believe this is a limitation of the fMRI dataset, not of the CLIP backbone or our learning-free distillation module. We have added figures on page 20 (section A.2 of the supplemental).
>
> * We consistently observe that CLIP dense features when probed with the CLIP text encoder can have very fine-grained semantic attributions, able to localize arms/legs/hair/eyes/ears/nose/mouth and other body parts for humans & animals, as well as clothing components.
>
> * However, when probed with fMRI voxel-wise weights or region of interest weights, we do not see consistent seperation between face and body voxels. We believe this is due to the fMRI stimulus set. Going over NSD images, we observe that faces are overwhelmingly presented together with a body. While our approach of freezing the fMRI encoder backbone helps mitigate the issue of co-occurring objects compared to training the whole network as done in brain dissection **[2]**, it cannot fully avoid this issue.
>
> > **Q3) Applying voxel-wise encoder to dense features**
>
> Your description is correct. For a given voxel, we train a linear probe to go from image embedding of shape $1\times M$ to a $1\times 1$ scalar brain response using a $M\times 1$ weight and $1\times 1$ bias. During dense inference we freeze the voxel-wise weights. We have dense features $H\times W \times M$, and we apply the voxels-wise weights to every patch independently. Unlike BrainSCUBA, we are not limited to linear probes, however we use them here as they empirically achieve good brain prediction performance.
>
> > **Q4) Learning-Free distillation and equations 3-5**
>
> Your description is correct. We were specifically motivated by Feature Field Distillation **[3]** by Kobayashi et al. Which proposed to learn a 3D semantic field via multi-view consistency (photometric loss). In 2D, we simulate multiple views via augmentation of the images (horizontal flips and shifts).
>
> The original feature field distillation required very expensive gradient-based training of a neural network using MSE loss for each scene (hours per-scene) and expensive inference procedures. This high computational cost during training and inference does not facilitate explorations over the visual cortex using tens of thousands of images. In equations 3 to 5, we show that aggregation via averaging achieves the same mathematical solution as MSE loss.
>
> We have added a discussion in page 19 (**section A.1**) of the revision, discussing the high level motivation behind learning-free distillation.
>
> > **Q5) Image-wise UMAPs**
>
> For each image, we compute a dense feature with learning-Free distillation. We then perform UMAP dimensionality reduction on the features for each image. The UMAP basis is not shared between images.
>
> The intention of this visualization was to partially answer questions similar to the one you raised about **body/face separation**.
>
> The figure shows that while the face and body components are indeed separated in the CLIP dense features, they are not separated in the fMRI saliencies. This is one of the reasons why we believe the lack of body/face separation is primarily from the fMRI stimulus/response, and not due to a failure of our method. We have added additional experiments in Page 20 (section A.2 of the supplemental) in response to your suggestion.
>
> > **Conclusion**
>
> We look forward to further discussion, and are happy to answer any questions! We hope our responses have been helpful, and that you could consider a more positive evaluation of our work.
>
> **References**
>
> **[1]** Selectivity for food in human ventral visual cortex (*Nature Communications Biology* 2023)
>
> **[2]** Brain dissection: fMRI-trained networks reveal spatial selectivity in the processing of natural images (*NeurIPS* 2023)
>
> **[3]** Decomposing NeRF for Editing via Feature Field Distillation (*NeurIPS* 2022)

---

> > ### Author Response · Authors · 2024-11-26
> >
> > Hello Reviewer yyM7,
> >
> > Do you have any remaining questions or suggestions after reviewing our response? Please feel free to share them with us!
> >
> > We are eager to answer any remaining questions and address your suggestions during the time remaining.

---

### Official Review · Reviewer_PZHq · 2024-11-09

**Soundness:** 3
**Presentation:** 3
**Contribution:** 3
**Rating:** 6
**Confidence:** 4

**Summary:**

The paper proposes a method, BrainSAIL, which aims to disentangle complex images into their semantically meaningful components and locate them to brain voxels or regions. With the prevalence of pretrained models (trained on very large amounts of data), deep learning offers a promising way to explore how semantics are organized in the brain cortex. Compared to prior methods, BrainSAIL focuses on selectivity in single-object images at the broad category level, thereby enabling a richer decomposition grounded in the full semantic complexity of natural visual experiences.

**Strengths:**

This is clearly written paper and makes a clear contribution. The idea of isolating specific visual features to determine selectivity effects in different cortical areas is novel and interesting. The proposed method can be used to explore the selectivity of higher visual cortex with respect to localized scene structure and image properties. This work achieves promising open vocabulary CLIP-based segmentation results.

**Weaknesses:**

The current experimental comparisons (open-vocabulary segmentation) are limited, which restricts a comprehensive evaluation of the proposed method. The proposed approach appears to closely resemble BrainSCUBA. Certain technical details are unclear, which makes it challenging to fully understand or reproduce the method.

**Questions:**

Was the same adapter used in Figures 1a and 1b? Also, were the adapter parameters frozen in Figure 1b? If they were frozen, why not use clean dense features to train the adapter?

How does the quality of the dense features impact the adapter's behavior? I suggest adding an ablation study to explore this effect.

---

> ### Author Response · Authors · 2024-11-21
> **Response to Reviewer PZHq [Part 1 of 2]**
>
> We thank Reviewer PZHq for the constructive review! We look forward to further discussion, and are happy to answer any questions. We have updated the paper to address your suggestions.
>
> > **Q1) Open-vocabulary segmentation evaluation**
>
> We follow SCLIP **[1]** & NACLIP **[2]** and perform quantitative evaluation on the four most widely used segmentation datasets -- `PASCAL VOC 2012`, `ADE20k`, `COCO-stuff`, and `COCO-Object` in ***Table 1*** of the main paper. We evaluate all three methods (SCLIP, NACLIP, and our method with learning-free distillation) without PAMR post-processing. The choice of no PAMR follows the suggestion of SCLIP (although we are compatible with PAMR), and we believe not using PAMR offers a more realistic evaluation of segmentation performance without assuming prior knowledge of the semantic classes or imposing a constraint on the spatial distribution of objects. We ran the segmentation code for all methods and consulted with the SCLIP and NACLIP authors on the evaluation procedure.
>
> We evaluate using two metrics:
>
> 1. The first metric is mean Intersection over Union (mIoU) -- and is a common metric in semantic segmentation. This metric assumes knowledge of the ground truth classes during evaluation in order to perform argmax over classes.
>
> 2. The second metric is the correlation of CLIP-based saliency maps against ground-truth labels. We compute saliency maps by computing the cosine similarity of CLIP-text embeddings against the dense features, and then measure pearson correlation of the saliency map against the ground truth one-hot labels. This metric is consistent with our goal of investigating the selectivity in the visual cortex *without* assuming knowledge of all the existing categories -- this is an important feature of our approach that distinguishes it from some prior work in which the categories are assumed *a priori*.
>
> NACLIP (WACV 2024) reported state-of-the-art open vocabulary segmentation results, and is itself an update of SCLIP (ECCV 2024). Our gradient-free distillation module consistently improves on NACLIP's results (as evaluated by mIoU and Pearson), which we independently verified as superior to SCLIP in most cases.
>
> The learning-free distillation module aggregation step takes under 0.2 seconds per image (NVIDIA L40S). Batched bf16 inference allows feature extraction for 51 views in under 0.5 seconds. All visualizations in our paper labeled `raw` are features from NACLIP. We have clarified this in the revised Figure 1 caption (see **blue text**).
>
> > **Q2) Comparison to BrainSCUBA**
>
> * **BrainSCUBA** constructs a method which outputs per-voxel natural language captions of voxel-wise selectivity.
>
>     * While this method is highly efficient, it outputs *a natural language descriptor*. Thus, selectivity is characterized through a natural language bottleneck, which means that the method may ignore semantic nuances that cannot be easily expressed through language. Moreover, existing captioning models struggle to capture these nuances effectively due to their limitations in representing non-verbal or context-dependent meanings.
>     * BrainSCUBA further forces the use of an encoder backbone which has a corresponding captioning network. In practice that means CLIP ViT-B/32 is the only model choice and the possible characterizations of voxels are restricted to this latent caption space.
>
> * **BrainSAIL** (our method), in contrast, outputs a spatial image-space localization using image-content and voxel-wise selectivity. Note that arbitrary combinations of voxels, as in fMRI regions of interest (ROIs) can also be used.
>
>     * Consequently, unlike BrainSCUBA, which outputs a natural language caption, **BrainSAIL** outputs *a relevency map over pixels (spatially distributed)*. As **BrainSAIL** *operates in image-space directly*, and does not go through a language bottleneck, it is far less constrained than BrainSCUBA (from a functional neural perspective).
>     * Furthermore, in contrast to BrainSCUBA, which was, in practice, restricted to CLIP ViT-B/32, **BrainSAIL** can be applied to any vision transformer (ViT) based model. This is illustrated in our paper, where we compare fMRI backbones from OpenAI (CLIP ViT-B/16), Google/Nvidia (SigLIP implementation by RADIOv2.5 ViT-B/16), and Facebook/Meta (DINOv2 ViT-B/14+reg). We find that despite overall similar test-time fMRI $R^2$, these networks actually rely on different parts of an image when used as an fMRI encoder. This was not something that could have been done using BrainSCUBA. Moreover, we view model comparisons (between models with different architectures, training objectives, etc.) as an important part of the work to better understand the neural computations realized in the brain **[3]**.
>
> **Response continued in next section**

---

> ### Author Response · Authors · 2024-11-21
> **Response to Reviewer PZHq [Part 2 of 2]**
>
> > **Q3) Technical details regarding Figure 1a and 1b**
>
> Regarding the voxel-wise adapter (orange in Figure 1), we indeed reuse that adapter from **1a** in **1b**. The weight/bias is frozen after obtaining it from  **1a**. It is computationally very difficult to train the encoder on the full dense embedding, as at native $224\times224$ resolution with $16$ patch size, that would require $14\times 14 = 196$ times more compute. It may be possible to train this with a spatially factorized encoder as done in **[4,5]**. However as we focus on higher visual cortex, the population receptive fields (pRF) are relatively large and invariant to a wide variety of 2D and 3D transformations **[6]**. Moreover, prior work **[3]** has found that these regions are well modeled by late layers of networks. We found that using the `[CLS]` token output gives us competitive test-set $R^2$ similar to **[3]**.
>
> The extraction of the dense features are done independently of the voxel-wise adapter (which is frozen). The voxel-wise adapter is not trained on dense features, and does not vary depending on the dense features. We have updated the paper to clarify this.
>
>
> **References**
>
> **[1]** SCLIP: Rethinking Self-Attention for Dense Vision-Language Inference (*ECCV* 2024)
>
> **[2]** Pay Attention to Your Neighbours: Training-Free Open-Vocabulary Semantic Segmentation (*WACV* 2024)
>
> **[3]** Better models of human high-level visual cortex emerge from natural language supervision with a large and diverse dataset (*Nature Machine Intelligence* 2023)
>
> **[4]** Generalization in data-driven models of primary visual cortex (ICLR 2021)
>
> **[5]** Neural system identification for large 579 populations separating “what” and “where.” (NeurIPS 2017)
>
> **[6]** Untangling invariant object recognition (*Trends in Cognitive Sciences* 2007)
>
> > **Conclusion**
>
> We hope that our clarifications have been helpful, and have updated our revision (highlighted in blue) according to your suggestions.

---

> ### Author Response · Authors · 2024-11-26
>
> Hello Reviewer PZHq,
>
> Do you have any remaining questions or suggestions following our response? Please let us know!
>
> We would be happy to assist in any way we can during the remaining time.

---

### Author Response · Authors · 2024-11-21
**General Response**

We are grateful to all reviewers for their constructive suggestions, which we agree will improve the clarity and communication of our work.

We are very encouraged by reviewers’ evaluation on the quality of this paper. All five reviewers find the work interesting -- "novel and interesting" (**PZHq**); "quite useful as an interpretation technique" (**yyM7**); "highlights potential equivalence between representations" (**PeSm**); provides "a more complete description of how real-world visual stimuli are represented and processed" (**dUkJ**); "method was explained clearly" (**PK4z**).

## General clarifications
### 1.1 Scope and experiments
* We propose BrainSAIL -- A method to localize regions in a image that a brain area is selective to. These spatial attribution maps utilize the dense embeddings from vision transformers.
* We leverage modifications to the self-attention mechanism that ensure the dense embeddings lie in the same embedding space as the summary token. We further denoise this dense representation using *learning-free distillation*. Inspired by multiview consistency in 3D neural field learning, we propose to generate 2D "views" with image augmentations.
* To mitigate the issue of slow neural field learning, we show mathematically that mean-aggregation achieves the same optimal representation as gradient-based MSE optimization. We find that our method consistently improves mIoU and correlation scores over NACLIP in **Table 1**, a method which claimed state of the art open-vocabulary segmentation results earlier this year.

We evaluate our method in four different ways, and leverage the ground truth category selectivity of voxels, measured using a functional localizer as part of that evaluation. We compare models that achieve similar prediction performance (CLIP, DINO, SigLIP), and find that they differ in terms of the features used.

* First, we perform dimensionality reduction on the higher visual cortex, and apply the same basis to image dense features.
    * We use UMAP applied to fMRI encoder weights and dense CLIP/DINO/SigLIP features.
    * We find that the image components align with known functionality in the visual cortex, including for people (EBA/FFA), food, and scenes (RSC/OPA/PPA)
* Second, we perform spatial attribution for regions of known functional selectivity.
    * We visualize the attributions, and find that they can function well even with brain regions selective to visually diverse objects like food.
    * We quantify the attributions by computing the category with the best spatial attribution alignment using the CLIP-text encoder for each brain region, and find that our method can always identify the correct top-category for face/body/place/food-selective voxels, but do not always agree in the word region (likely due to food and word region overlap in the brain).
* Third, we investigate low-level voxel-wise visual correlates using voxel-wise BrainSAIL. Across the visual cortex using depth, color saturation, and color luminance.
    * We find that depth correlates can identify scene regions.
    * We find the region surrounding FFA commonly identified to be food selective to be correlated with high color saturation
    * We find that OPA is divided into high/low color luminance, corresponding with previously identified outdoor/indoor selectivity.
* Finally, we compare the features utilized by different brain encoders. We investigate CLIP (vision-language contrastive), DINO (vision contrastive), and SigLIP (vision-language non-contrastive) models. While they achieve similar $R^2$, the models differ in terms of the features used for brain prediction.
    * We find that models that utilize language are more robust to visually diverse features that are part of the same object.
    * Vision only models are more sensitive and brittle to components that are visually different in color or shading.
    * Models differ the most for visually diverse objects (words and food), while agreeing on images which could be characterized by simple features (rectilinear features in scenes)

## 2. Updates and new experiments in the revision
1. We have clarified that our voxel-wise weights are frozen when used for inference on the dense features. (Page 3, **PZHq**)
2. We have added a discussion and citation on the potential limitations caused by using NSD (Page 10, **PeSm**, **dUkJ**)
4. We have added an experiment comparing the segmentation capabilities of CLIP and the brain (Page 20, **yyM7**, **PK4z**)
6. We have reorganized the supplemental and added additional information on fMRI data processing (Page 19, **PeSm**)
7. We better motivate our learning-free distillation, and clarify the purpose of equations 3 to 5. (Page 19, **yyM7**, **PeSm**)
8. We have added a previously missing axis label (Page 10, **PeSM**)

We genuinely appreciate the suggestions, and believe our paper will be improved with your feedback. Please let us know if you have any additional questions or comments!

---

### Meta-Review · Area_Chair_CwJ9 · 2024-12-18

**Metareview:**

This paper proposes a method, named BrainSAIL, which aims at investigating the semantic topology of the human brain by analyzing fMRI scans as correlated to semantic concepts in natural images. It leverages dense visual features and whole image embeddings extracted by a deep learning pre-trained model to identify specific sub-regions of each image, reflecting the selectivity patterns in different areas of the higher visual cortex.

This work was mainly appreciated since the beginning (6, 6, 8, 6, 5), lately becoming unanimously positive (6, 6, 8, 6, 6) after rebuttal.

Main significant, valuable aspects concern the clear contribution of the designed task and the approach to cope with it, the clarity of the presentation, and the principled experimentation analysis.

There are also some issues, mainly related to the novelty, some confusing parts in the method's explanation, some weak aspects of the experimental analysis, as related to comparisons and the presentation of the results, and a request of further results as considering fine-grained categories. Other punctual comments have also been raised by a couple of reviewers.

Overall, authors provided extensive and accurate comments to all raised remarks, which were appreciated by the reviewers who maintained or raise their scores.

In conclusion, this paper is recommended for acceptance to ICLR 2025.

**Additional Comments On Reviewer Discussion:**

See above

---

### Decision · Program_Chairs · 2025-01-22

Accept (Poster)